# Entropy and crystal-facet modulation of P2-type layered cathodes for long-lasting sodium-based batteries

Fang Fu[1,7 ✉], Xiang Liu[2,7], Xiaoguang Fu[1,7], Hongwei Chen[1], Ling Huang[3], Jingjing Fan[3], Jiabo Le[3], Qiuxiang Wang[1], Weihua Yang[1], Yang Ren[4], Khalil Amine[2,5,6 ✉], Shi-Gang Sun[3 ✉] & Gui-Liang Xu[2 ✉]

P2-type sodium manganese-rich layered oxides are promising cathode candidates for sodium-based batteries because of their appealing cost-effective and capacity features. However, the structural distortion and cationic rearrangement induced by irreversible phase transition and anionic redox reaction at high cell voltage (i.e., >4.0 V) cause sluggish Na-ion kinetics and severe capacity decay. To circumvent these issues, here, we report a strategy to develop P2-type layered cathodes via configurational entropy and ion-diffusion structural tuning. In situ synchrotron X-ray diffraction combined with electrochemical kinetic tests and microstructural characterizations reveal that the entropy-tuned $Na_{0.62}Mn_{0.67}Ni_{0.23}Cu_{0.05}Mg_{0.07}Ti_{0.01}O_2$ (CuMgTi-571) cathode possesses more {010} active facet, improved structural and thermal stability and faster anionic redox kinetics compared to $Na_{0.62}Mn_{0.67}Ni_{0.37}O_2$. When tested in combination with a Na metal anode and a non-aqueous $NaClO_4$-based electrolyte solution in coin cell configuration, the CuMgTi-571-based positive electrode enables an 87% capacity retention after 500 cycles at 120 mA g$^{-1}$ and about 75% capacity retention after 2000 cycles at 1.2 A g$^{-1}$.

[1] College of Materials Science and Engineering, Huaqiao University, Xiamen 361021, China. [2] Chemical Sciences and Engineering Division, Argonne National Laboratory, Lemont, IL 60439, United States. [3] State Key Laboratory of Physical Chemistry of Solid Surfaces, College of Chemistry and Chemical Engineering, Xiamen University, Xiamen 361005, China. [4] X-ray Science Division, Advanced Photon Source, Argonne National Laboratory, Lemont, IL 60439, United States. [5] Materials Science and Engineering, Stanford University, Stanford, CA, United States. [6] Materials Science and Nano-engineering, Mohammed VI Polytechnic University, Lot 660 Hay Moulay Rachid, Ben Guerir, Morocco. [7] These authors contributed equally: Fang Fu, Xiang Liu, Xiaoguang Fu. ✉email: fufang@hqu.edu.cn; amine@anl.gov; sgsun@xmu.edu.cn; xug@anl.gov

Na-ion batteries (NIBs) have attracted significant interest as a low-cost alternative to Li-ion systems for large-scale energy storage applications due to the natural abundance of Na resources and the similar chemical properties of Na and Li[1–3]. The foremost challenge in advancing NIBs lies in developing high-performance and low-cost electrode materials. Among the available cathode materials, P2-type layered oxides, especially sodium manganese-rich oxides ($Na_xMnO_2$) have been deemed as one of the most fascinating cathode candidates due to their high specific capacities and the natural abundance of Mn[4–6]. Moreover, oxygen anionic redox activity has been confirmed in these cathodes when charging beyond 4.0 V (vs. Na/Na$^+$), which can increase the reversible capacity to even over 200 mAh g$^{-1}$ at low charge/discharge specific current (e.g., 10 mA g$^{-1}$)[5,7,8]. However, compared with Li$^+$, the insertion/extraction of Na$^+$ in the reported P2 cathodes generally shows poor reversibility and sluggish kinetics owing to its larger ionic radius (1.02 Å for Na$^+$ vs. 0.76 Å for Li$^+$, Supplementary Table 1). Moreover, shuttling the large and heavy Na$^+$ leads to the gliding of $MO_2$ layers and the formation of the irreversible O2 or OP4 phase accompanied with the pronounced volume variation during cycling, leading to fast capacity decay and short cycle life[5,8,9]. On the other hand, the irreversible lattice oxygen due to the structural instability results in detrimental cationic migration and oxygen network distortion, leading to voltage decay and poor kinetics[10,11].

Over the past decades, extensive research has been conducted to stabilize anionic redox in the Mn-based P2-type layered cathodes[12–14]. However, the reported cycle stability by surface coating or aliovalent doping, especially under fast-charge conditions remain far less satisfied[15–19]. Instead, rather less attention has been paid to improving ion transport kinetics. Ion transport kinetics within electrode materials play a pivotal role in the battery performance, which depends highly on the host structure and ion-diffusion tunnels. A stable host structure can avoid the collapse of lattice and diffusion channels, which is the precondition for reversible ion insertion/extraction. The construction of more diffusion-favorable tunnels is beneficial for the rapid migration of large amounts of ions. Thus, the key to addressing the slow ion kinetics lies in constructing a stable host structure with more migration tunnels.

Recently, high-entropy oxides have shown significantly improved structural stability during charge/discharge[20,21]. Hu et al. has revealed that high-entropy O3-type $NaNi_{0.12}Cu_{0.12}Mg_{0.12}Fe_{0.15}Co_{0.15}Mn_{0.1}Ti_{0.1}Sn_{0.1}Sb_{0.04}O_2$ cathode could enable reversible O3-P3 phase transition and cationic redox within a narrower voltage range of 2.0–3.9 V[20]. In addition to structural stability, {010} planes with an open structure and ion-migration tunnels are favorable active planes for ion transport of layered cathode materials. Our previous work has demonstrated that increasing {010} active planes could significantly improve the rate performance of lithium layered oxide cathodes[22,23]. P2 cathodes with increased exposure of {010} facets have also been reported, which however were evaluated only within a low-voltage region (2.0–4.0 V).

High-voltage operation (>4.2 V) is an effective way to increase the specific capacity of battery materials. However, it is well-known in the battery community that high-voltage operation often leads to undesired structural degradation of the cathode. To date, despite numerous attempts through various strategies, the stabilization of layered cathodes at high voltage remains a formidable challenge. To the best of our knowledge, the correlation between entropy and anionic redox kinetics/reversibility of layered cathodes at high voltage remains elusive.

Based on the above discussions, simultaneously increasing the entropy and active facets would be a powerful gateway to enable P2-type layered cathodes with ultrafast charging capability and ultralong cycle life (Fig. 1), which however has not been reported so far. In this work, we have designed and synthesized a series of multi-element P2-type $Na_{0.62}Mn_{0.67}Ni_{0.23}Cu_{0.05}Mg_{0.09-2y}Ti_yO_2$ (CuMgTi-571, y = 0.01; CuMgTi-552, y = 0.02; CuMgTi-533, y = 0.03) layered cathodes with different configurational entropy and {010} active facets. Through systematically comparing their electrochemical performance and investigating their phase transition and Na$^+$ diffusion coefficient during charge/discharge, we have identified the correlation between entropy and active crystal facet, cationic/anionic kinetics as well as an electrochemical performance of P2-type cathodes. In comparison with $Na_{0.62}Mn_{0.67}Ni_{0.37}O_2$, the optimal $Na_{0.62}Mn_{0.67}Ni_{0.23}Cu_{0.05}Mg_{0.07}Ti_{0.01}O_2$ cathode with higher entropy and increased active {010} facet not only effectively suppressed destructive structural variation and extends cycle life, but also significantly promoted the anionic redox and fast charging capability during high-voltage cycling (i.e., 2.0–4.3 V cell voltage range).

## Results

**Structural and morphological characterization.** The crystalline structures and phase purities of the entropy-tuned $Na_{0.62}Mn_{0.67}Ni_{0.23}Cu_{0.05}Mg_{0.09-2y}Ti_yO_2$ and the non-entropy-tuned $NaMNO_2$ were characterized by X-ray diffraction (XRD) together with the Rietveld refinement. As shown in Fig. 2, all the characteristic peaks from each sample are sharp, clear, and well fitted with the typical hexagonal P2-structure with P63/mmc space group, indicating the high purity and good crystallinity of the samples[24,25]. The pure P2 phase of $Na_{0.62}Mn_{0.67}Ni_{0.23}Cu_{0.05}Mg_{0.09-2y}Ti_yO_2$ suggests uniform incorporation of Cu$^{2+}$, Mg$^{2+}$, and Ti$^{4+}$ into the lattice of the materials. Compared to Mn$^{4+}$ (0.53 Å) and Ni$^{2+}$ (0.69 Å), the introduction of larger-size cations such as Cu$^{2+}$ (0.73 Å) and Mg$^{2+}$ (0.72 Å) may lead to a debate on the size variation of interlayer spacing. However, previous literature has shown that interlayer spacing doesn't always increase with an increased substitution amount of large-sized cations[16]. Meanwhile, a high entropy sample $Na_{0.62}Mn_{0.67}Ni_{0.23}Co_{0.033}Al_{0.033}Ti_{0.02}O_2$ consisting of small-sized cations (Co$^{3+}$: 0.61 Å, Al$^{3+}$: 0.53 Å, Ti$^{4+}$: 0.60 Å) was synthesized and showed that introduction of cations with smaller ionic radii into

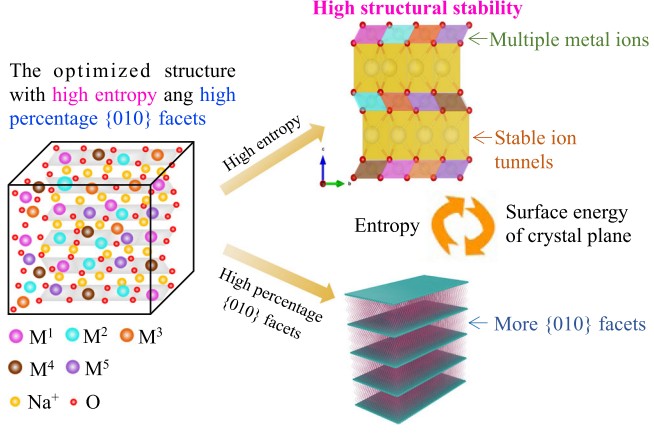

**Fig. 1 Design strategy of optimized P2-type cathode structure.** High-entropy layered cathodes with multiple cations could provide high structural stability and stable ion-diffusion tunnels. A high percentage of {010} facets in the layered cathodes could offer more channels for fast ion migration. In addition, entropy variation may affect the energy of the system and crystal planes.

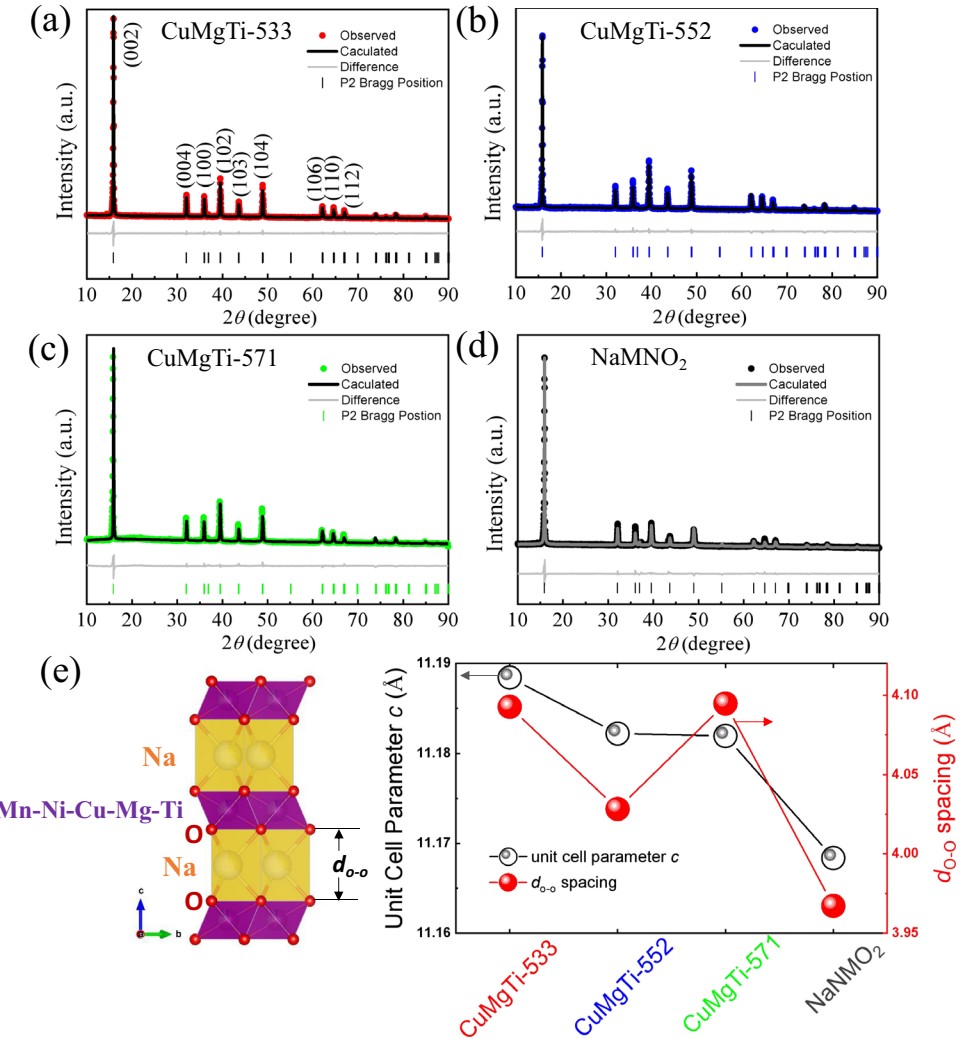

**Fig. 2 Interlayer spacing of synthesized cathodes.** XRD patterns and Rietveld refinement plots of **a** CuMgTi-533, **b** CuMgTi-552, **c** CuMgTi-571, and **d** NaMNO$_2$ samples. **e** The detailed atomic model of interlayer spacing ($d$o-o) in P2-type layered structure is presented on the left. The values of lattice parameter $c$ and interlayer spacing for four samples are presented on the right.

P2-typed structure can lead to increased interlayer spacing (Supplementary Fig. 1).

Therefore, to obtain the lattice parameters and interlayer spacing of Na$_{0.62}$Mn$_{0.67}$Ni$_{0.23}$Cu$_{0.05}$Mg$_{0.09-2y}$Ti$_y$O$_2$ and NaMNO$_2$ samples, the synthesized materials were refined by the Rietveld method as displayed in Fig. 2a–d, respectively. According to the refined crystallographic data presented in Fig. 2e and Supplementary Tables 2–5, the Mn$^{4+}$, Ni$^{2+}$, Cu$^{2+}$, Mg$^{2+}$, and Ti$^{4+}$ ions are located in the octahedral 2a Wyckoff sites. Na$^+$ ions are located in two prismatic sites (2b and 2d, respectively). The interlayer spacings ($d$o-o) of CuMgTi-533, CuMgTi-552, CuMgTi-571, and NaMNO$_2$ were 4.09, 4.03, 4.09, and 3.98 Å, respectively. The maximum changes of interlayer distance ($d$) are 0.11 Å. Compared to previous report work by increasing the Na$^+$ layer distance from 3.6 to 5.8 Å to improve the electrochemical performance[26], the interlayer distance change here is much smaller.

The morphologies and detailed crystal structures of the entropy-tuned Na$_{0.62}$Mn$_{0.67}$Ni$_{0.23}$Cu$_{0.05}$Mg$_{0.09-2y}$Ti$_y$O$_2$ and the non-entropy-tuned NaMNO$_2$ were characterized by scanning electron microscopy (SEM) and transmission electron microscopy (TEM). Low-magnification SEM images (Supplementary Fig. 2) showed that all the as-prepared samples are composed of micrometer-sized particles with predominantly 1–2 μm in diameter.

High-magnification SEM images (Fig. 3a–d) revealed that these particles have rather smooth surfaces over the whole particle. TEM images from the individual structure (Supplementary Fig. 3a–d) further confirmed the smooth and clean surfaces of these micrometric particles. The atomic compositions of Na$_{0.62}$Mn$_{0.67}$Ni$_{0.23}$Cu$_{0.05}$Mg$_{0.09-2y}$Ti$_y$O$_2$ and NaMNO$_2$ samples were confirmed by inductively coupled plasma-mass spectrometry (ICP-MS), as displayed in Supplementary Table 6. The ICP results revealed that the atomic ratios of metals in all samples are highly consistent with the designated atomic ratio. Moreover, as shown in Fig. 3e and Supplementary Fig. 4, the elemental mappings by high angle annular dark-field scanning transmission electron microscopy (HAADF-STEM) confirmed the existence and uniform distribution of Na, Mn, Ni, Cu, Mg, and Ti in the Na$_{0.62}$Mn$_{0.67}$Ni$_{0.23}$Cu$_{0.05}$Mg$_{0.09-2y}$Ti$_y$O$_2$, and Na, Mn, Ni in the NaMNO$_2$ samples.

The selected area electron diffraction (SAED) patterns of CuMgTi-571 (Fig. 3f) exhibited hexagonal spot patterns, which is an indication of the single-crystalline feature and hexagonal crystalline structure of the microparticles. Based on the hexagonal phase system of P2-type layered oxides, it can be inferred that the front facets of microparticles are {001} planes and the side facets are {010} planes (Fig. 3g). The atomic arrangement of {010} planes presents an open structure, which facilitates Na$^+$ transport

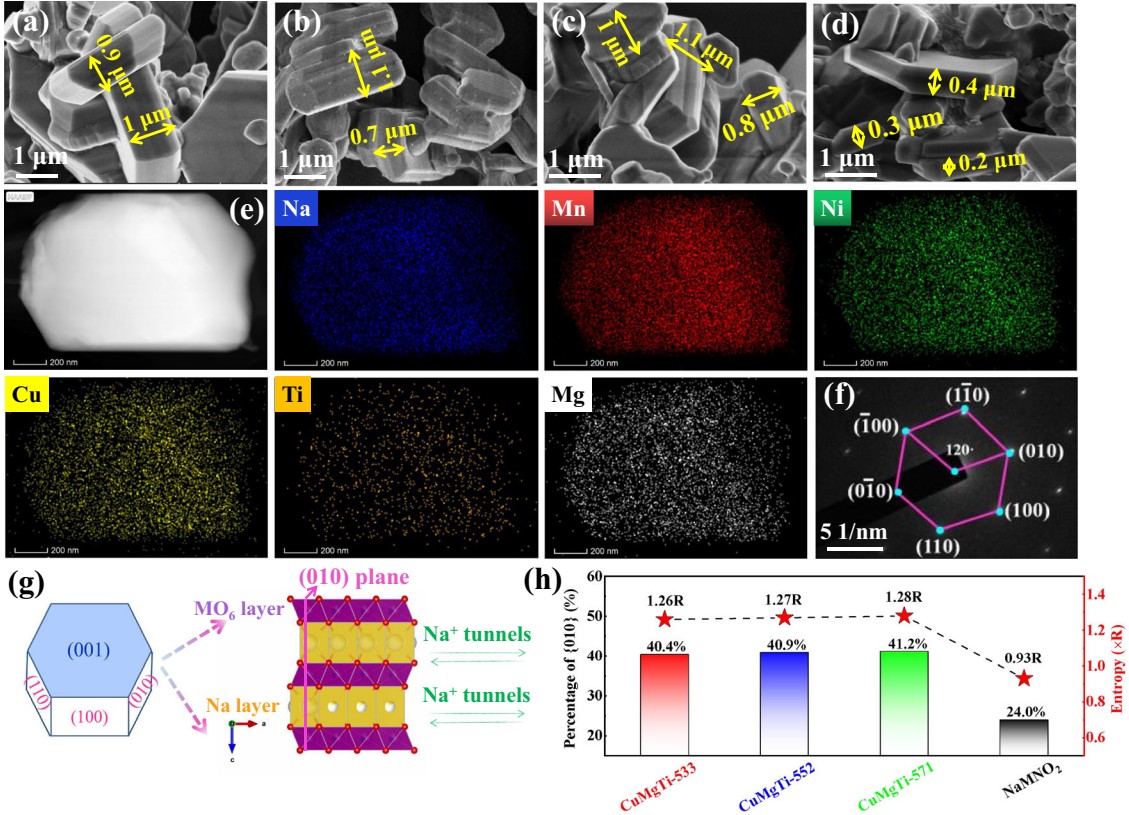

**Fig. 3 Physicochemical characterizations of the cathodes.** SEM images of **a** CuMgTi-533, **b** CuMgTi-552, **c** CuMgTi-571, and **d** NaMNO$_2$ samples.
**e** HAADF-STEM images and the corresponding EDX elemental mappings of Na, Mn, Ni, Cu, Mg, and Ti in the CuMgTi-571. **f** SAED pattern of CuMgTi-571.
**g** Schematic illustration of microparticles with six {010} facets and two {001} facets. The surface atomic arrangement of (010) facet is shown on the right.
**h** The percentage of {010} facets and configurational entropy of CuMgTi-533, CuMgTi-552, CuMgTi-571, and NaMNO$_2$ samples.

between the MO$_2$ layers (Fig. 3g). Surprisingly, it can be clearly observed that the side surfaces of the Na$_{0.62}$Mn$_{0.67}$Ni$_{0.23}$-Cu$_{0.05}$Mg$_{0.09-2y}$Ti$_y$O$_2$ hexagonal-prism particles are generally thicker than that of the NaMNO$_2$ hexagonal-prism particles. The average thickness of the CuMgTi-533, CuMgTi-552, and CuMgTi-571 hexagonal-prism particles were measured to be 0.88, 0.90, and 0.91 μm, respectively, around 2.2 times thicker than that of NaMNO$_2$ (0.41 μm) (Supplementary Fig. 2e–h). By assuming the edge length of the hexagonal front facets was 1.5 μm, the percentages of {010} facets in the CuMgTi-533, CuMgTi-552, and CuMgTi-571 particles were calculated to be 40.4, 40.9, and 41.2%, respectively, which are all higher than that of NaNMO$_2$ (24.9%) (Fig. 3h). The increased surface area of {010} facets in the entropy-tuned Na$_{0.62}$Mn$_{0.67}$Ni$_{0.23}$Cu$_{0.05}$Mg$_{0.09-2y}$Ti$_y$O$_2$ could provide more efficient and direct pathways for Na$^+$ transport.

The configurational entropy ($S_{config}$) of Na$_{0.62}$Mn$_{0.67}$Ni$_{0.23}$-Cu$_{0.05}$Mg$_{0.09-2y}$Ti$_y$O$_2$ materials depends on the number and mole fraction of the elements, which can be calculated according to Eq. (1)[27]:

$$S_{config} = -R\left[\left(\sum_{i=1}^{N} x_i \ln x_i\right)_{\text{cation-site}} + \left(\sum_{j=1}^{M} x_j \ln x_j\right)_{\text{anion-site}}\right]$$ (1)

where $x_i$ and $x_j$ represent the mole fraction of elements present in the cation and anion sites, respectively, and $R$ is the universal gas constant. The contribution of the anion-site has a minor influence on the $S_{config}$. Thus, the effect of anion is not considered (Supplementary Note 1). As shown in Fig. 3h, the configuration entropy of CuMgTi-533, CuMgTi-552, CuMgTi-571, and NaMNO$_2$ were 1.26 R, 1.27 R, 1.28 R, and 0.93 R, respectively,

which showed the same change tendency as the percentage of {010} facets.

Based on the calculation formula of surface energy (Eq. (2)):

$$\gamma(T) = \gamma_0 - (T/A)(S_{vib} + S_{config})$$ (2)

Where $\gamma(T)$ and $\gamma_0$ are surface energy at any temperature and 0 K, respectively. $T$ is temperature, $A$ is Helmholtz free energy, $S_{vib}$ and $S_{config}$ are the system's vibrational and configurational entropy, respectively. It can be found that the surface energy of a system with simple composition and low entropy is mainly influenced by temperature, while the surface energy of polycomponent system with high entropy is mainly influenced by configurational entropy $S_{config}$[28]. The P2-type layered cathode materials in our current work contain six cations, which are high-entropy systems. The surface energy of these P2-type entropy-tuned materials is mainly affected by configurational entropy $S_{config}$ at a given temperature. Thus, configurational entropy variation may reasonably alter the surface energy of the crystal plane, influencing the growth rate of different facets and promoting the formation of the layered cathode with more {010} facets. Therefore, high configurational entropy is the impetus for the growth of a high percentage {010} active facets in this work, while the temperature is the driving force for the growth of {010} active facets in the case of O3 lithium compounds (LiNi$_{1/3}$Co$_{1/3}$Mn$_{1/3}$O$_2$ and Li(Li$_{0.17}$Ni$_{0.25}$Mn$_{0.58}$)O$_2$) with low entropy[22,23]. These results indicate that (configurational) entropy variation may reasonably alter the surface energy of crystal planes influencing the growth rate of different facets[29]. To the best of our knowledge, the effect of configurational entropy on the surface structure of electrode materials has been barely reported and requires more comprehensive study.

**Electrochemical performances**. To clearly unravel the effect of entropy and crystal-facet modulation on the structural stability and performance of P2-structured cathode material, a series of electrochemical characterizations were performed. Cyclic voltammetry (CV) measurements were conducted to compare the redox behaviors of the entropy-tuned $Na_{0.62}Mn_{0.67}Ni_{0.23}$-$Cu_{0.05}Mg_{0.09-2y}Ti_yO_2$ and the non-entropy-tuned $NaMNO_2$. As shown in Supplementary Fig. 5, similar to most P2-type cathode materials, both $Na_{0.62}Mn_{0.67}Ni_{0.23}Cu_{0.05}Mg_{0.09-2y}Ti_yO_2$ and $NaMNO_2$ cathodes exhibit multiple pairs of redox peaks, corresponding to a series of phase transition reactions occurring during charge/discharge. According to the previously reported results, the low-voltage reduction peak located at ca. 2.26 V is associated with the reduction reaction of $Mn^{4+}/Mn^{3+}$ (Supplementary Fig. 5)[30,31]. Other weak peaks in 2.0–3.0 V are associated with the $Na^+$/vacancy ordering process. The redox peaks at ca. 3.47/3.30 and 3.70/3.52 V could be attributed to the redox processes of $Ni^{2+}/Ni^{3+}$ and $Ni^{3+}/Ni^{4+}$, respectively (Supplementary Fig. 5)[31]. The high-voltage redox peaks at ca. 4.29/3.96 V are related to oxygen anionic redox, which can be demonstrated by the results of X-ray photoelectron spectroscopy (XPS) analysis (Supplementary Fig. 6). Compared with the CV plots of the $NaMNO_2$ cathode, a new pair of extraction/insertion peaks at ca. 3.95/3.81 V appears in all the $Na_{0.62}Mn_{0.67}Ni_{0.23}Cu_{0.05}Mg_{0.09-2y}Ti_yO_2$ cathodes and could be attributed to the redox behavior of $Cu^{2+}/Cu^{3+}$ (Supplementary Fig. 5)[32]. The CV results suggest that $Ni^{2+}/Ni^{4+}$ and $Cu^{2+}/Cu^{3+}$ as well as $(O^{2-}/O_2^{n-})$ $(1 \le n \le 3)$ are the redox-active couples within the potential range applied and will contribute to the capacity of the as-prepared cathodes.

The anionic redox reaction $(O^{2-}/O_2^{n-})$ $(1 \le n \le 3)$ is reversible in the first CV curves of CuMgTi-571 and $NaMNO_2$ (Supplementary Fig. 5). However, they show a significant difference in the voltage hysteresis (e.g., the oxidation and reduction peak of anionic redox locates at different potentials). The hysteresis phenomenon is often observed in the cathode materials with anionic redox[33–36]. The peak potential differences ($\Delta E_P$) of anionic redox reaction in $NaMNO_2$ is 0.59 V (4.30 and 3.71 V for oxidation and reduction peak, respectively). In contrast, the ($\Delta E_P$) of anionic redox reaction in CuMgTi-571 is decreased to 0.33V (4.29 and 3.96 V for oxidation and reduction peak, respectively).

Figure 4a, b shows the charge/discharge profiles of CuMgTi-571 and $NaMNO_2$ at 0.1 C (12 mA g$^{-1}$) of different cycles (both materials were tested using a Na metal anode in coin cell configuration), respectively. It can be seen that the $NaMNO_2$ undergoes a severe voltage fade and further specific capacity decay during cycling, which are characteristic features of Li-excess cathodes that also involve oxygen anionic redox[37,38]. By contrast, the CuMgTi-571 cathode exhibits less voltage fading during the cycling test in comparison with the $NaMNO_2$ cathode, indicating that the voltage fade issue in CuMgTi-571 is alleviated. This result can be seen more clearly in the normalized voltage capacity curve in Supplementary Fig. 7a, b. Figure 4c compares the cycle performance of CuMgTi-571 with higher entropy and $NaMNO_2$ with much lower entropy at 0.1 C (12 mA g$^{-1}$). As shown, CuMgTi-571 could deliver an initial discharge capacity of 148.2 mAh g$^{-1}$, and can still maintain 132.9 mAh g$^{-1}$ after 100 cycles, resulting in capacity retention of 89.6%. Although $NaMNO_2$ delivers a comparable first discharge capacity (150.5 mAh g$^{-1}$) with CuMgTi-571, the discharge capacity of $NaMNO_2$ decreases fast and remains only 60.2% (90.6 mAh g$^{-1}$) of its initial capacity after 50 cycles. Moreover, the average coulombic efficiency of CuMgTi-571 achieves 98.0% compared to 95.4% of $NaMNO_2$, suggesting the fast dynamics and high reversibility of CuMgTi-571. To investigate the reversibility of electrochemical reaction in CuMgTi-571 and $NaMNO_2$, dQ/dV curves after 50 cycles at 0.1 C (12 mA g$^{-1}$) rate are displayed in

Supplementary Fig. 8. It can be seen that the anionic redox peaks at high voltage have been well maintained in the CuMgTi-571 in comparison with $NaMNO_2$. These sharp contrasts confirmed that entropy modulation can significantly stabilize the structures and anionic redox of P2 layered cathodes during high-voltage charge/discharge.

The rate capability of the entropy-tuned $Na_{0.62}Mn_{0.67}Ni_{0.23}$-$Cu_{0.05}Mg_{0.09-2y}Ti_yO_2$ and the non-entropy-tuned $NaMNO_2$ were further evaluated at various charge/discharge rates from 0.1 C to 10 C (both materials are tested using a Na metal anode in coin cell configuration, 1 C = 120 mA g$^{-1}$, Fig. 4d and Supplementary Fig. 9). The discharge capacity of $NaMNO_2$ decreases dramatically with increasing C-rates, while all the $Na_{0.62}Mn_{0.67}Ni_{0.23}$-$Cu_{0.05}Mg_{0.09-2y}Ti_yO_2$ shows better rate performance than $NaMNO_2$. Among them, CuMgTi-571 demonstrates the best rate capability. When the rates are increased successively from 0.1 to 0.2, 0.5, 1, 2, 5 and 10 C (1 C = 120 mA g$^{-1}$), the CuMgTi-571 cathode enables cell-specific capacities of 143.5, 128.3, 122.9, 120.1, 115.1, 104.4, and 82.6 mAh g$^{-1}$, respectively. When the C-rate switches back to 0.1 C (12 mA g$^{-1}$), a discharge capacity of 151.8 mAh g$^{-1}$ can be immediately recovered, indicating the tolerance for fast Na$^+$ transport and stable structure of CuMgTi-571 cathode. The changes in the interlayer distance ($d$) between NaNMO$_2$ and CuMgTi-571 are only 0.11 Å. Such a small variation of interlayer distance is not sufficient to account for the improvement in the rate capability. In contrast, the relatively larger entropy difference (e.g., 0.35 R between CuMgTi-571 and $NaMNO_2$) and active-facet percentage difference (e.g., 16.3% between CuMgTi-571 and $NaMNO_2$) should play a dominant role in the observed improvement on rate capability. With an increase in the entropy and active facet, the reversible capacities of $Na_{0.62}Mn_{0.67}Ni_{0.23}Cu_{0.05}Mg_{0.09-2y}Ti_yO_2$ under various rates were increased accordingly.

A similar trend can be found in their long-term cyclability, which is an important parameter for battery materials when used for practical applications. In this regard, their long cycling performances were further evaluated by charging/discharging at 1 C (120 mA g$^{-1}$) for 500 cycles (The materials are tested using a Na metal anode in coin cell configuration, Fig. 5). Figure 5a, b show the voltage profiles of CuMgTi-571 and $NaMNO_2$ for the 100th, 200th, 300th, 400th, and 500th cycles during charge/discharge at 1 C (120 mA g$^{-1}$) rate, respectively. CuMgTi-571 cathode again displays less voltage fading during the long cycling compared with the $NaMNO_2$ cathode. Figure 5c compares the cycling performances of $Na_{0.62}Mn_{0.67}Ni_{0.23}Cu_{0.05}Mg_{0.09-2y}Ti_yO_2$ and $NaMNO_2$ at 1 C (120 mA g$^{-1}$) for 500 cycles. It can be clearly seen that the CuMgTi-571 cathode with the highest entropy and {010} active facet percentage also displays better long-term stability with the highest reversible capacity (103.3 mAh g$^{-1}$) and capacity retention (87%) after 500 cycles. While the reversible capacities and retention of CuMgTi-552 (89.8 mAh g$^{-1}$, 82.7%) and CuMgTi-533 (65.9 mAh g$^{-1}$, 62.7%) were decreased accordingly. In sharp contrast, the $NaMNO_2$ with the lowest entropy and least {010} active facet exhibits rapid capacity decay, which can only maintain a reversible capacity of 30.9 mAh g$^{-1}$ after 500 cycles. In addition, the $NaMNO_2$ cathode also exhibits a lower Coulombic efficiency than the CuMgTi-571 cathode during charge/discharge (Supplementary Fig. 10a).

Furthermore, we tested the long-term cycle stability of CuMgTi-571 and $NaMNO_2$ cathode under extreme fast charging/discharging of 10 C (ca. 6 min charge/discharge, 10 C = 1200 mA g$^{-1}$, both materials are tested using a Na metal anode in coin cell configuration). The discharge capacity of the $NaMNO_2$ cathode faded rapidly during cycling and dropped to 22.9 mAh g$^{-1}$ at the 1000th cycle (Fig. 5d). The CuMgTi-571 cathode can still display a fast-charging performance with a reversible capacity of 59.3 mAh g$^{-1}$

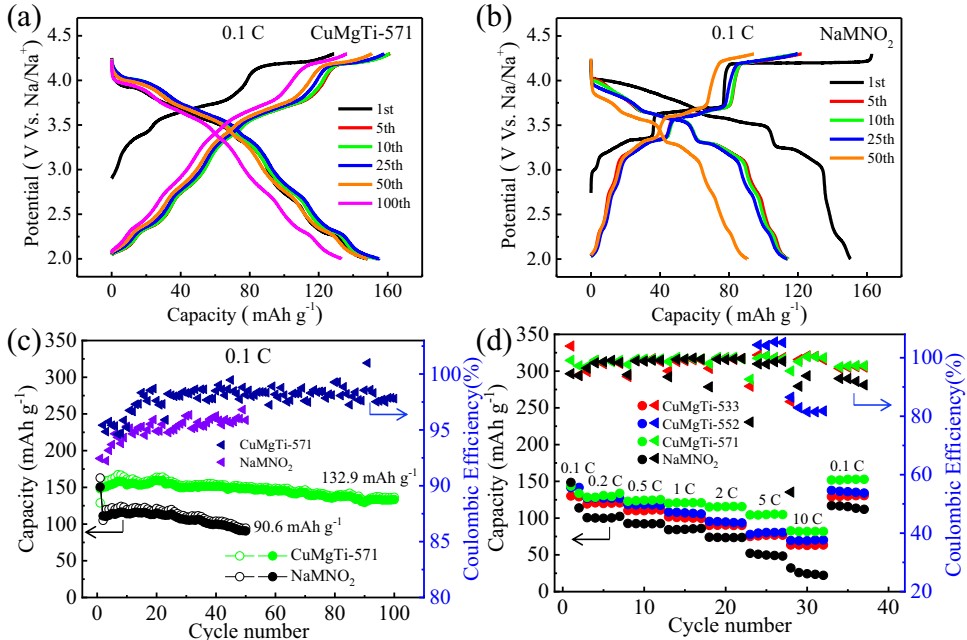

**Fig. 4 Cycling and rate performances of $Na_{0.62}Mn_{0.67}Ni_{0.23}Cu_{0.05}Mg_{0.09-2y}Ti_yO_2$ and $NaMNO_2$ cathodes.** Charge/discharge profiles of **a** CuMgTi-571 and **b** $NaMNO_2$ at 0.1 C (12 mA g$^{-1}$) between 2.0 and 4.3 V. **c** Cycling performances of CuMgTi-571 and $NaMNO_2$ at 0.1 C (12 mA g$^{-1}$) for 100 cycles. **d** Rate performances of $Na_{0.62}Mn_{0.67}Ni_{0.23}Cu_{0.05}Mg_{0.09-2y}Ti_yO_2$ and $NaMNO_2$ from 0.1 C to 10 C (1 C = 120 mA g$^{-1}$). Cycling and rate tests were conducted in a coin cell using Na metal as an anode at 25 °C. Hollow and solid circles in (**c**, **d**) represent charge and discharge capacity, respectively. Triangles represent coulombic efficiency.

and reasonable capacity retention of 75.4% after 2000 cycles (Fig. 5d). The Coulombic efficiency of CuMgTi-571 cathode is closer to 100% in comparison with that of $NaMNO_2$ cathode, indicating better sodiation/de-sodiation reversibility (Supplementary Fig. 10b). We compared the electrochemical performance of our entropy and active-facet tuned cathode with other reported P2-type layered cathodes tested at different conditions (Supplementary Table 7 tested at a high rate and Supplementary Table 8 tested at the same cut-off voltage). The comparison indicates a significant advance of the CuMgTi-571 in terms of capacity, cycle number, capacity retention even high-rate capability.

To further verify the benefit of entropy-driven structural stability on the Na$^+$ kinetics of the P2-type layered cathodes, $Na_{0.62}Mn_{0.67}Ni_{0.21}Cu_{0.05}Mg_{0.016}Zn_{0.016}Sn_{0.016}Y_{0.016}O_2$ and $Na_{0.62}Mn_{0.67}Ni_{0.21}Cu_{0.05}Mg_{0.015}Zn_{0.015}Sn_{0.015}Zr_{0.015}O_2$ with eight cations and higher entropy were prepared and tested (both materials are tested using a Na metal anode in coin cell configuration, Supplementary Fig. 11). These eight cations' high-entropy materials still exhibit good long-term cycling stability and rate performance (e.g., >64.5% capacity retention for 3000 cycles at 10 C (1200 mA g$^{-1}$), Supplementary Fig. 12a) as well as high Coulombic efficiency (Supplementary Fig. 12b). The results confirm the structural stability and fast ion kinetics of the layered cathodes achieved by increasing the configurational entropy of the system.

The Na$^+$ diffusion coefficients ($D_{Na+}$) of CuMgTi-571 and $NaMNO_2$ cathode were measured through the galvanostatic intermittent titration technique (GITT), and the detailed calculation process is described in Supplementary Note 2. As shown in Fig. 6a, b, the charge and discharge processes were divided into four regions based on the different electrochemical behaviors for better interpretation. Figure 6c–f show the average $D_{Na+}$ of CuMgTi-571 and $NaMNO_2$ in four voltage regions. The average $D_{Na+}$ for both samples are around 10$^{-8}$ cm$^2$ s$^{-1}$ in the low and middle voltage region (2.0–4.05 V) during either charging or

discharging, where the charge compensation is mainly contributed by the cationic redox (Ni$^{2+}$/Ni$^{3+}$, Mn$^{3+}$/Mn$^{4+}$, and Cu$^{2+}$/Cu$^{3+}$). In contrast, the average $D_{Na+}$ for both samples are around 10$^{-11}$ cm$^2$ s$^{-1}$ in the high-voltage range of 4.05–4.3 V, where the charge compensation is mainly contributed by the anionic redox (O$^{2-}$/O$_2^{n-}$) (1 ≤ n ≤ 3). The results indicate that anionic redox has much slower kinetics than cationic redox, which is the rate-determining step. Thus, enhancing the kinetics of anionic redox is crucial for the performance improvement of Mn-based P2-type layered cathodes. From Fig. 6d, e, it can be clearly observed that the $D_{Na+}$ of CuMgTi-571 at high voltage is higher than those of $NaMNO_2$. During 4.05–4.3 V, the average $D_{Na+}$ of CuMgTi-571 is ~10 times higher than that for $NaMNO_2$, suggesting the kinetics of anionic redox is greatly improved in the entropy and crystal-facet tuned cathode.

To examine the effect of the conductivity on the anionic redox reaction of CuMgTi-571 and $NaMNO_2$ at high voltage, in situ electrochemical impedance spectroscopy (EIS) experiments were conducted using Na metal as a counter/reference electrode. Supplementary Fig. 13 shows the Nyquist plots of the CuMgTi-571 and $NaMNO_2$ at different potentials of the first charge and discharge process. Upon Na$^+$ insertion/extraction, both samples experience a series of processes in the bulk and at the surface, leading to different suppressed semicircles in Nyquist plots of the first charge and discharge process[39]. At the high voltage region (4.0–4.3 V), the Nyquist plots of both samples exhibit two distinct parts including a depressed semicircle in the high-frequency region and an inclined line in the low-frequency zone. The high-frequency semicircle should be related to the charge-transfer resistance[39–41]. By comparing the diameters of the semicircles in the high-voltage region (Supplementary Table 9), the impedances of CuMgTi-571 and $NaMNO_2$ are similar, implying similar electronic conductivity of the two de-sodiated samples. Therefore, the significantly improved kinetics of the anionic redox in the

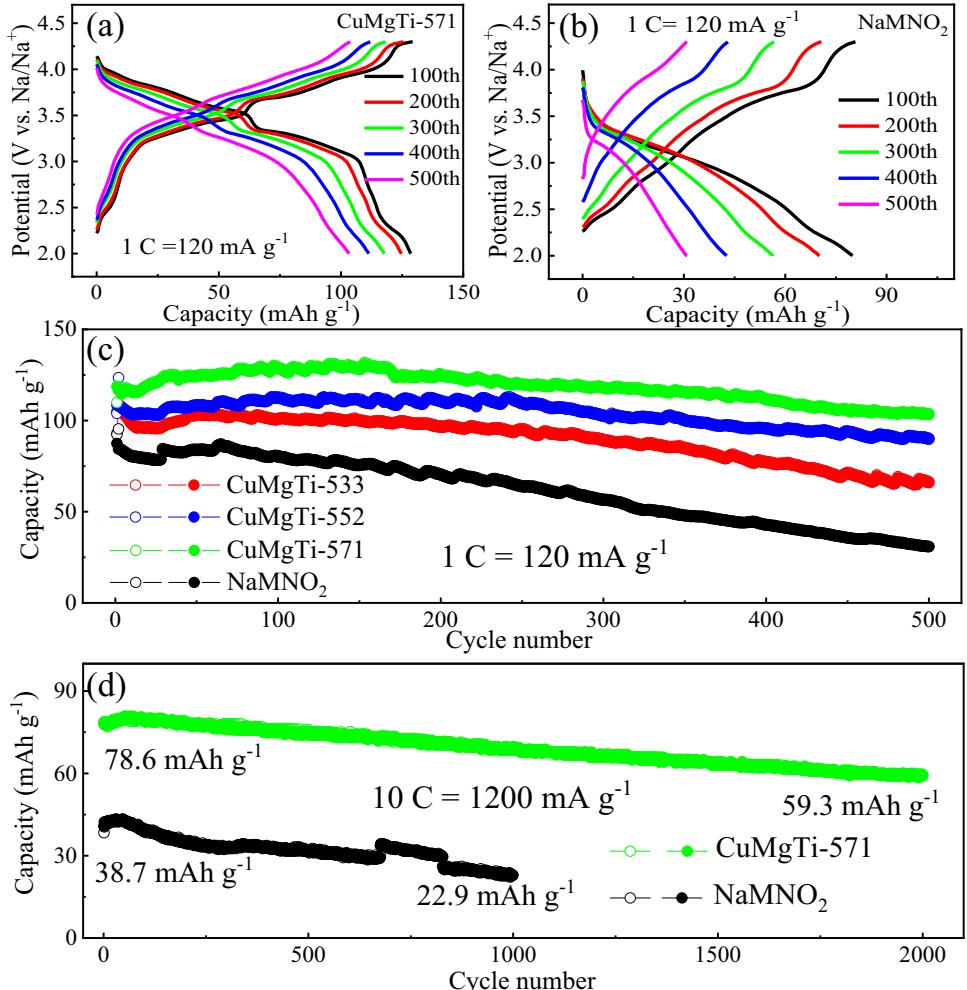

**Fig. 5 Long cycling performances of $Na_{0.62}Mn_{0.67}Ni_{0.23}Cu_{0.05}Mg_{0.09-2y}Ti_yO_2$ and $NaMNO_2$ cathodes.** Charge/discharge profiles of **a** CuMgTi-571 and **b** $NaMNO_2$ at 1 C (120 mA g$^{-1}$) between 2.0 and 4.3 V. **c** Cycling performances of $Na_{0.62}Mn_{0.67}Ni_{0.23}Cu_{0.05}Mg_{0.09-2y}Ti_yO_2$ and $NaMNO_2$ at 1.0 C (120 mA g$^{-1}$) for 500 cycles; **d** Long-term cycling behavior of CuMgTi-571 and $NaMNO_2$ at high rate of 10 C (1200 mA g$^{-1}$). Long cycling tests were conducted in coin cells using Na metal as an anode at 25 °C. Hollow and solid circles in (**c**, **d**) represent charge and discharge capacity, respectively.

CuMgTi-571 is mainly contributed by its structure with high active facets and high entropy, which can provide more stable migration tunnels and is helpful for accelerating Na$^+$ diffusion.

**Phase transition and structural stability.** Na$^+$ kinetics is strongly associated with the structural stability of the material and anionic redox usually results in the generation of oxygen vacancies and migration of transition metal (TM) ions, which could cause phase transition and new phase formation. To check the impact of entropy and crystal-facet tuning on the structural stability of the material, in situ synchrotron high energy X-ray diffraction (HEXRD) was conducted on CuMgTi-571 during charge/discharge (tested in coin cell using a Na metal anode), and the results are displayed in Fig. 7 and Supplementary Fig. 14. Figure 7a, b shows the waterfall and contour plot of HEXRD patterns of the CuMgTi-571 electrode charged/discharged at 0.1C (12 mA g$^{-1}$) between 2.0 and 4.3 V, respectively. Upon Na$^+$ extraction (charge), the (002) and (004) peaks shift toward a lower 2$\theta$° angle due to the increase in the repulsive electrostatic interaction between adjacent MO$_2$ sheets; while (100), (102), (108), and (112) peaks consecutively peaks shift toward a higher 2$\theta$° angle with the contraction of a-, b-axis[42–44]. During the subsequent discharge, the XRD pattern of CuMgTi-571 exhibited an exactly opposite evolution. During the whole charge/discharge process, the

well-defined peaks corresponding to the P2 phase are well maintained and returned to the initial states after one full cycle. No new peaks belonging to OP4[45], O2[8], or Z[6] phase are detected when charged to 4.3 V, and no existing peaks vanish or split, indicating that new phase formation and phase transition do not occur in this material. The changes in lattice parameters upon cycling based on the refinement results are displayed in Fig. 7c. It is clear that the lattice parameter evolution is highly reversible during the charge and discharge process. The largest changes of the *a*, *c*, and *V* in the first cycle are only 0.74, 0.91, and 0.57%, respectively, which are much smaller than those of the state-of-the-art reported layered cathodes (Supplementary Table 10 and Supplementary Note 3). In addition, the structural stability of CuMgTi-571 at different cycles was also investigated and the results are shown in Supplementary Fig. 15. The XRD patterns of the cycled samples and pristine are almost identical, indicating that the P2-type structure can be well maintained after extensive cycling. The high reversibility of lattice parameter evolution and structural evolution upon Na$^+$ (de-) intercalation manifests the unusual structure stability of entropy and crystal facet tuned cathode.

In situ HEXRD during heating of de-sodiated cathode powder was also performed to investigate the phase stability of entropy and crystal-facet tuned cathodes at high temperatures (heating from 30 to 450 °C) to investigate their thermal tolerance

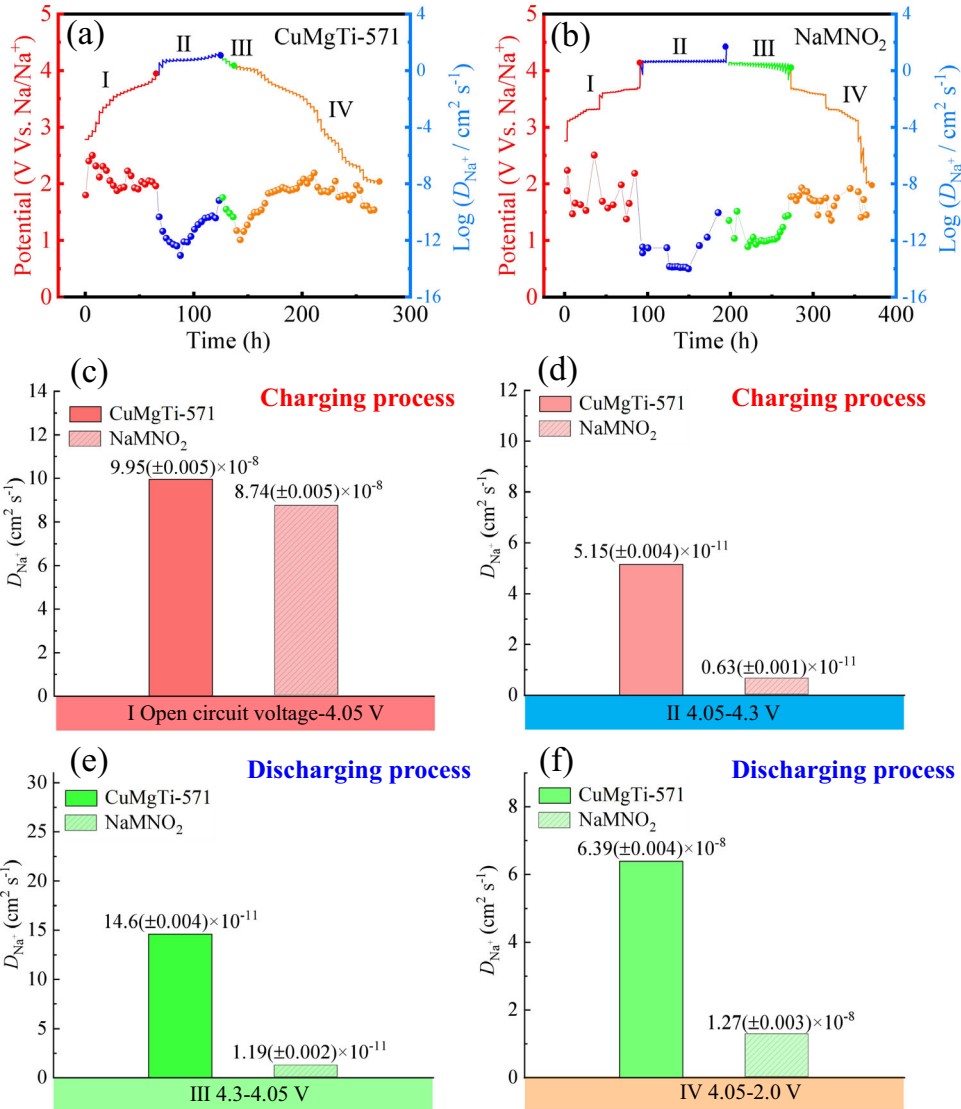

**Fig. 6 Investigations on the Na-ion diffusion coefficient for different cathode materials.** GITT curves and corresponding $Na^+$ diffusion coefficients of **a** CuMgTi-571 and **b** NaMNO$_2$ during the charge and discharge processes. **c–f** The average $Na^+$ diffusion coefficient of CuMgTi-571 and NaMNO$_2$ in different voltage regions. GITT tests were performed in coin cells using Na metal as an anode at 25 °C.

(Fig. 8a, b). The tested CuMgTi-571 and NaMNO$_2$ materials were scraped from 4.2 V-charged cathodes. The (002) peak, which is most sensitive to the Na content during charging/discharging, is usually used to monitor the phase transformation of P2-type layered cathodes during heating. The (002) reflection of charged CuMgTi-571 continuously moves to a lower angle with increasing the temperature due to the lattice expansion caused by heating (Fig. 8a). Distinct from CuMgTi-571, the (002) reflection of charged NaMNO$_2$ first shifts to a lower angle upon heating, and then moves to a higher angle when the heating temperature reached above 222 °C (Fig. 8b). Such phenomenon indicates that new phases such as spinel or rock-salt were formed in the charged NaMNO$_2$ during the heating. Moreover, the (002) peak fading starts at 350 and 322 °C for the CuMgTi-571 and NaMNO$_2$ cathodes, respectively, further confirming the inferior structural stability of the NaMNO$_2$. The formation of a new phase and inferior structural stability would lead to irregular migration channels and hence sluggish Na$^+$ diffusion.

To further unravel the robust structural stability of the entropy and crystal-facet tuned cathodes, the morphologies of CuMgTi-571 and NaMNO$_2$ after 500 cycles at 1 C (1 C = 120 mA g$^{-1}$) were

further compared (both materials using a Na metal anode are tested in coin cell configuration). SEM images in Fig. 8c and Supplementary Fig. 16a show a well-preserved morphology of the CuMgTi-571 particles without noticeable structural degradation, displaying a stable inner structure robust enough to endure the long-term insertion/extraction of Na$^+$. In contrast, the morphology of the NaMNO$_2$ particles changed after long-term cycling (Fig. 8d and Supplementary Fig. 16b). The smooth side surfaces before the cycles split into many thin sheets after 500 cycles, indicating that the repeated Na$^+$ de/intercalation leads to volumetric expansion and serious structural degradation, which could be associated with the fast capacity fade of the NaMNO$_2$.

## Discussion

The above results clearly demonstrate that fast and reversible (de-) intercalation of large Na$^+$ have been achieved simultaneously in the Na$_{0.62}$Mn$_{0.67}$Ni$_{0.23}$Cu$_{0.05}$Mg$_{0.09-2y}$Ti$_y$O$_2$. The Na$^+$ extraction/insertion kinetics are mainly attributed to the increased structural stability and ion-diffusion channels. Configurational entropy contribution to the decreased Gibbs free energy in materials retards

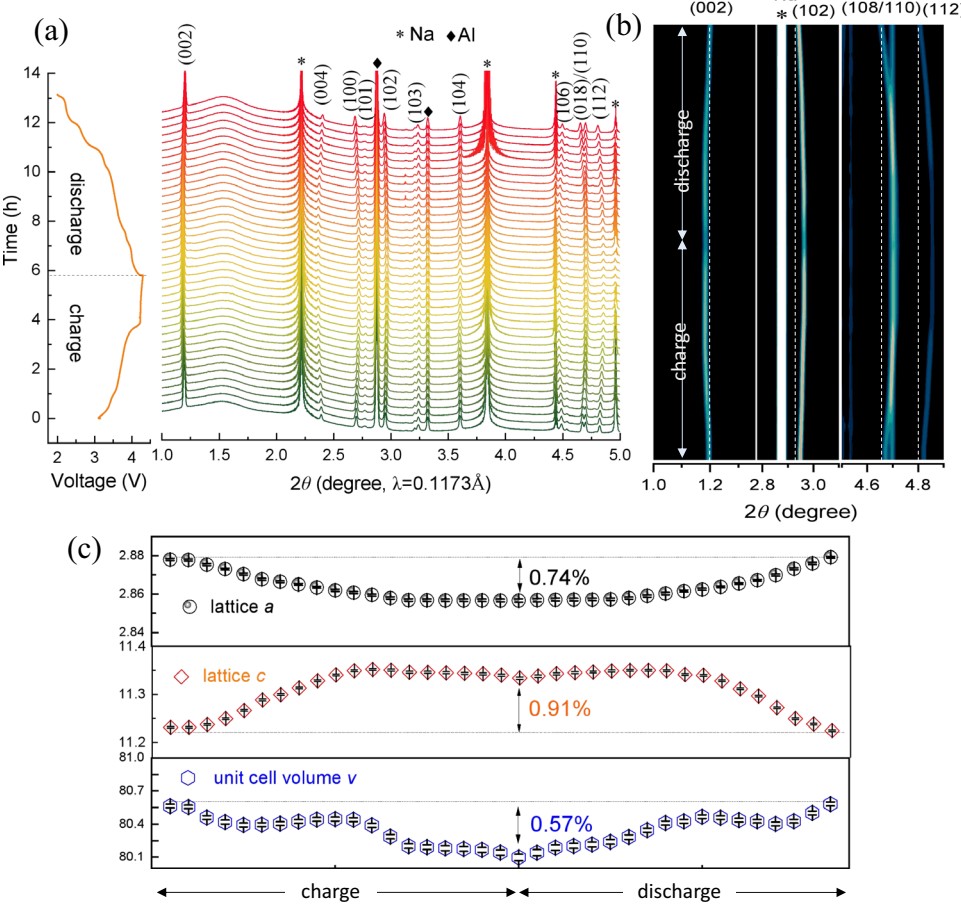

**Fig. 7 In situ HEXRD characterization in coin cell configuration at 25 °C. a** The waterfall plot and **b** contour plot of in situ HEXRD patterns for CuMgTi-571 within 2.0–4.3 V. **c** Evolution of cell parameters and cell volume during the charge/discharge process.

dislocation movement and stabilizes crystal structure. Such entropy-driven high structural stability is robust enough to restrain volume variation, structure distortion, and irreversible phase transition, thus safeguarding a stable migration path and reversible $Na^+$ (de-) intercalation. Moreover, the increased {010} electrochemically active planes due to the increased entropy create more favorable and efficient tunnels for the migration of $Na^+$, which facilitate rapid Na$^+$ (de-)intercalation and promote electrochemical kinetics.

In summary, configurational entropy could result in a thermodynamically stable structure through a local minimization of Gibbs free energy, and a high percentage of {010} facets in the layered cathodes could provide more channels for ion transport. Based on the advantages of high entropy and high active facets, a combinatorial strategy of entropy modulation and active-facet modulation was developed to stabilize the structure and optimize ion-migration pathways of P2-type layered cathodes. The correlation of entropy, active facet, $Na^+$ kinetics, and electrochemical performance of the $Na_{0.62}Mn_{0.67}Ni_{0.23}Cu_{0.05}Mg_{0.09-2y}Ti_yO_2$ was systematically studied by a combination of TEM, electrochemical measurement, in situ XRD, in situ EIS and GITT. The results demonstrate that increased entropy and active facet can effectively stabilize crystal structure and accelerate electrochemical kinetics of sluggish anionic redox, thus enabling the fast and reversible migration of $Na^+$. In virtue of entropy-driven structural stability and increased migration channels, an optimal CuMgTi-571 cathode with higher entropy and a more active facet shows fast charging/discharging capability up to 10 C (~6 min) and long-term durability for over 2000 cycles.

## Methods

**Materials synthesis**. A series of multi-element P2-structured layered oxides ($Na_{0.62}Mn_{0.67}Ni_{0.23}Cu_{0.05}Mg_{0.09-2y}Ti_yO_2$, y = 0.01, 0.02, and 0.03) were prepared using a stirring hydrothermal method. In a typical synthesis, stoichiometric amounts (with 7% excess sodium) of $NaCH_3COO\cdot3H_2O$, $Mn(CH_3COO)_2\cdot4H_2O$, $Ni(CH_3COO)_2\cdot4H_2O$, $Cu(CH_3COO)_2\cdot H_2O$, $Mg(CH_3COO)_2\cdot4H_2O$, and $TiO_2$ were mixed in 60 mL of deionized water. An appropriate amount of $H_2C_2O_4$ (molar ratio of oxalic acid to metal ions is 1.3:1) was used as a precipitant agent and also dissolved in deionized water. $H_2C_2O_4$ solution was added dropwise into the above mixture under stirring. Afterward, the reaction mixture was put into 100 mL Teflon-lined autoclaves and aged at 130 °C for 12 h under stirring. The green precipitates were collected by solvent evaporation at 80 °C. The resulting precipitates were annealed in air at 500 °C for 6 h and then at 900 °C for 12 h, and P2-$Na_{0.62}Mn_{0.67}Ni_{0.23}Cu_{0.05}Mg_{0.09-2y}Ti_yO_2$ samples were finally obtained. In addition, $Na_{0.62}Mn_{0.67}Ni_{0.37}O_2$, $Na_{0.62}Mn_{0.67}Ni_{0.21}Cu_{0.05}Mg_{0.016}Zn_{0.016}Sn_{0.016}Y_{0.016}O_2$, and $Na_{0.62}Mn_{0.67}Ni_{0.21}Cu_{0.05}Mg_{0.015}Zn_{0.015}Sn_{0.015}Zr_{0.015}O_2$ were also prepared by same method. The obtained P2-type layered $Na_{0.62}Mn_{0.67}Ni_{0.23}Cu_{0.05}Mg_{0.03}Ti_{0.03}O_2$, $Na_{0.62}Mn_{0.67}Ni_{0.23}Cu_{0.05}Mg_{0.05}Ti_{0.02}O_2$, $Na_{0.62}Mn_{0.67}Ni_{0.23}Cu_{0.05}Mg_{0.07}Ti_{0.01}O_2$, and $Na_{0.62}Mn_{0.67}Ni_{0.37}O_2$ samples were labeled as CuMgTi-533, CuMgTi-552, CuMgTi-571, and NaMNO$_2$, respectively.

**Physicochemical characterizations**. All the samples were characterized by scanning electron microscopy (SEM, Hitachi S-4800), transmission electron microscopy (TEM, JEOLJEM-2100), inductively coupled plasma-mass spectrometry (ICP-MS, Thermo X2), X-ray diffraction (XRD, Bruker D8 Discovers X-ray analytical systems with Cu Kα radiation), energy dispersive X-ray (EDX, an EDX detector system attached to FEI Talos-S), and X-ray photoelectron spectroscopy (XPS, ESCALAB 250Xi).

The cycled electrode samples for ex situ XPS and XRD characterization were obtained by disassembling the cycled half-cells using Na metal anode in Ar-filled glovebox ($H_2O < 0.1$ ppm, $O_2 < 0.1$ ppm). The electrodes were rinsed using dimethyl carbonate (DMC, 99%, Sigma Aldrich) repeatedly, then dried in glovebox to remove the solvents. For ex situ XPS measurements, the dried electrodes were first placed into a vacuum transfer vessel in glovebox and then transferred into the

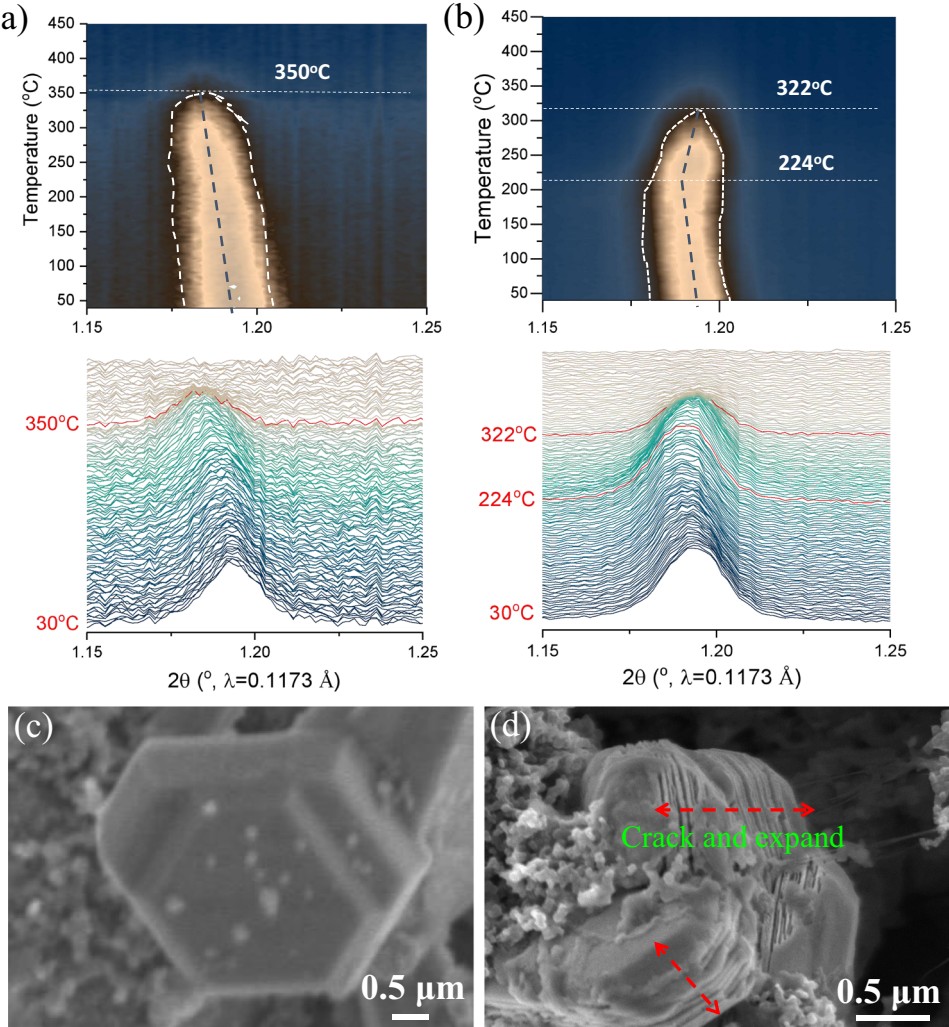

**Fig. 8 Structural and morphological investigations of harvested cathode materials.** The contour plots and waterfall plots of de-sodiated **a** CuMgTi-571 and **b** NaMNO$_2$ during the heating from 30 °C to 450 °C at 5 °C min$^{-1}$, the tested samples during heating are de-sodiated cathode powder scraped from 4.2 V-charged cathodes. SEM images of the **c** CuMgTi-571 and **d** NaMNO$_2$ cathodes in the discharged state of 2.0 V after 500 cycles at 1 C (120 mA g$^{-1}$).

XPS ultrahigh vacuum chamber. For ex situ XRD measurements, the dried electrodes were transferred from glovebox and then tested in air.

**HEXRD measurements**. The in situ synchrotron high energy XRD characterizations of the CuMgTi-571 electrode were carried out at the beamline 11-ID-C of the Advanced Photon Source at Argonne National Laboratory with a wavelength of 0.1173 Å. The Na|CuMgTi-571 coin cells (type CR2032) with two identical holes on the top and bottom cases were assembled in the argon-filled glovebox and then sealed with Kapton tape after filling with the electrolyte. The electrolyte was 1 M NaPF$_6$ (≥98%, Sigma Aldrich) dissolved in propylene carbonate (PC, 99%, Sigma Aldrich) with 2 vol% fluoroethylene carbonate (FEC, 99%, Sigma Aldrich). A constant charge/discharge current of 20 mA g$^{-1}$ was applied by using a MACCOR system between 2.0 and 4.3 V at 25 °C. Each 2D diffraction pattern was obtained by a PerkinElmer amorphous silicon 2D detector during the charge/discharge process and then were converted to 1D patterns using fit2D software with a calibrated using a standard CeO$_2$ sample with the same experimental setup. The postmortem cathode HEXRD measurements during the heating was also conducted at 11-ID-C of Advanced Photon Source. Both the NaNMO$_2$ and NiMgTi-571 cathode were charged to 4.2 V and then disassembled in the glovebox and scratch off the active materials carefully. The charged cathode materials were then loaded into a stainless-steel DSC autoclave and sealed in the glovebox to prevent any humid/air contamination during the in situ heating experiment. A Linkam TS1500 furnace was used to heat the charged samples from 30 to 450 °C at 5 °C min$^{-1}$ and a PerkinElmer 2D detector was continuously used to acquire the 2D diffraction data during the heating process.

**Electrochemical characterization**. All the electrochemical measurements were performed using CR2016 coin cells assembled in an Ar-filled glovebox. Cathodes were prepared by casting a slurry composed of 70 wt% active material, 20 wt% acetylene black, and 10 wt% polyvinylidene difluoride (PVDF, HSV900, Arkema) binder dissolved in N-methyl-2-pyrrolidone (NMP, 99.9%, Aladdin) on an Al foil (99%, 16 μm in thickness, Hefei Kejing, China) and drying the slurry under vacuum at 120 °C for 12 h. The areal loading of the active material was about 3–4 mg cm$^{-2}$. A high-purity Na metal foil was used as anode and glass fiber as separator (675 μm in thickness, 2.7 μm average pore size, Whatman). Na metal foil (1.5 mm in thickness, 99.5%, Sinopharm) was made from Na chunks under mineral oil, which was removed from the surface of Na by dried paper inside the glovebox. Na chunk was then pressed into a thin sheet under 50 Pa inside the glovebox, which was further punched into 16 mm diameter plates and used as an anode. The electrolyte was 1 M NaClO$_4$ dissolved in a mixed solvent of propylene carbonate (PC) and fluoroethylene carbonate (FEC) (49:1, v/v). The amount of electrolyte used in the coin cells was controlled at around 280 μL. Charge-discharge experiments were performed galvanostatically between 2.0 and 4.3 V at different C-rates (1 C = 120 mA g$^{-1}$) on a LANHE-CT2001A test system (Wuhan, China). Both in situ XRD cells were charged and discharge under 2.0–4.3 V. Cyclic voltammetry (CV) measurements were carried out on a CHI760E electrochemical workstation (Chenhua, Shanghai China) at a scanning rate of 0.2 mV s$^{-1}$ within the potential of 2.0–4.3 V. Galvanostatic intermittent titration technique (GITT) tests were conducted in the voltage range of 2.0–4.3 V by applying repeated current pulses at a current rate of 0.1 C for 15 min followed by a 3 h rest. The in situ electrochemical impedance spectroscopy (EIS) experiments were executed using a PARSTAT2273 electrochemical workstation in a frequency range from 0.1 Hz to 100 KHz with an AC amplitude of 5 mV. EIS tests of Na||CuMgTi-571 and Na||NaMNO$_2$ coin cells

were performed after each potentiostatic intermittent titration technique (PITT) program completed. The step potential difference was 0.1 V, and the step elapsed time was 3600 s. Each EIS test contains 50 data points. All the electrochemical tests were made at 25 °C.

**Reporting summary**. Further information on research design is available in the Nature Research Reporting Summary linked to this article.

## Data availability
The data that support the findings of this study are available from the corresponding authors F.F., K.A., S.-G.S and G.-L.X. upon reasonable request.

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

## Acknowledgements
This work is supported by the National Natural Science Foundation of China (Grant No. 21805100), National Key Research and Development of China (2016YFB0100202), the Natural Science Foundation of Fujian Province (No. 2019J05091 and 2019J06018), and the Promotion Program for Young and Middle-aged Teacher in Science and Technology Research of Huaqiao University (ZQN-PY606). Research at Argonne National Laboratory was funded by the US Department of Energy, Vehicle Technologies Office under contract no. DE-AC02-06CH11357. Use of the Advanced Photon Source, an Office of Science User Facility operated for the DOE Office of Science by Argonne National Laboratory, was supported by DOE under contract no. DE-AC02-06CH11357. K.A. and G.-L.X. also thank the support from Clean Vehicles, US-China Clean Energy Research Centre (CERC-CVC2).

## Author contributions
F.F. and G.-L.X. conceived the idea and designed the experiments; X.F. prepared the P2-structured layered cathodes and conducted the electrochemical measurements; X.L. performed in situ HEXRD experiments and processed the data; H.C., L.H., J.F., J.L., Q.W., W.Y., and Y.R. helped with data analysis. F.F. and G.-L.X. wrote and revised the

manuscript; F.F., G.-L.X., S.-G.S., and K.A. managed the project; all authors contributed to discussions and paper revisions.

## Competing interests

The authors declare no competing interests.
