## [Peer Review File · Nature Communications]

Reviewers' comments:

Reviewer #1 (Remarks to the Author):

Revision Manuscript ID: NCOMMS-330337_0_art_file_5880920

Title: Entropy and crystal-facet modulation enable ultrafast and ultralong-life P2-type layered cathodes for Na-ion batteries

The manuscript reports on a novel strategy to develop ultrafast and ultra-stable P2-type layered cathodes by simultaneously and systematically regulating their configurational entropy and ion-diffusion tunnels.

In situ synchrotron X-ray diffraction combined with electrochemical kinetic tests and microstructural characterization collectively revealed that the entropy-tuned $\text{Na}_{0.62}\text{Mn}_{0.67}\text{Ni}_{0.23}\text{Cu}_{0.05}\text{Mg}_{0.07}\text{Ti}_{0.01}\text{O}_2$ (CuMgTi-571) cathode possesses more {010} active facet, better structural stability and thermal stability as well as 10-times faster anionic redox kinetics than that of $\text{Na}_{0.62}\text{Mn}_{0.67}\text{Ni}_{0.37}\text{O}_2$. Authors claim that their entropy and crystal facet modulation open up strategies to obtain high power and long-life sodium batteries. In their work they "have identified the hidden correlation between entropy and active crystal facet, cationic/anionic kinetics as well as electrochemical performance of P2-type cathodes".

The overall study is well presented. However, the manuscript lacks in novelty and while the main outcome might be presented as highly innovative the main results confirm studies and observations already seen in the field. For this main reason it is suggested to submit the manuscript to a different journal. Some comments and questions highlighting these points are of the reasons are explained below:

1. High entropy cathode materials for SIBs have been already proposed by Hu et al. (Zhao, C. et al. High-Entropy Layered Oxide Cathodes for Sodium-Ion Batteries. *Angew. Chem. Int. Ed.* 59, 264-269 (2020). More generally, high entropy oxides (HEOs), represented by the multi-element metallic systems that can crystallize in a single phase, is a known strategy. In this work the novel aspect could be identified with the fact that authors combine high entropy materials with active crystal facets. However, also this strategy has already been proposed. The same authors of this manuscript have reported on the same strategy on O3 type lithium compounds (ref 23 and 24 of this manuscript). The novelty is not very clear, beside the fact that this time the strategy is being applied to Na-based P2-type materials. In addition, considering how the percentage of {010} facets have been calculated (see page 7 and Fig. 3), it appears that the main difference among the investigated samples is associated to an increase of the interlayer distance ($d=11.16 \text{ \AA}$ vs 11.18 \AA for NM and CuMgTi-533, CuMgTi-552, CuMgTi-571 respectively), which certainly is beneficial for an improved kinetic of sodiation and desodiation enhancing rate capability. Overall, this phenomenon has already been comprehensively investigated and discussed in literature. The data are certainly well presented and represent a good piece of work, which however can be classified as incremental and does not represent an innovative breakthrough in the field. As such, publication in Nature Comm is not recommended.

2. Authors claim that: "their results indicate that entropy variation may alter the surface energy of crystal planes, and further influences the growth rate of different facets. To the best of our knowledge, the effect of the entropy on the growth rate of crystal planes have not been exploited and required more comprehensive study." The high entropy and the active crystal facets are a consequence of the introduction in the materials of multiple metal ions with larger ionic radii [larger ionic sizes of Cu^{2+} (0.73 \AA) and Mg^{2+} (0.72 \AA) compared to Mn^{4+} (0.53 \AA) and Ni^{2+} (0.69 \AA)]. This induces not only a higher entropy but also a larger interlayer distance into the layered structure, suggesting that the

main effect on the improved rate capability is associated only to a larger interlayer distance. As a matter of fact, authors claim that "increasing configuration entropy may be beneficial for improving the growth rate of layered oxide cathodes along the [001] direction and promote the formation of layered cathode with more {010} facets".

What would have happened if the increased entropy would have been achieved by using different metals, for instance with smaller ionic radii?

To evaluate this, author should be able to prepare sample with high entropy by using smaller ionic radii substituents. The comparison will elucidate the differences. Would then the materials still have active crystal facets?

3. Table 6 in Supp Info provides calculated entropy value for literature data. It would be interesting to see if the d spacing of the materials reported present the same trend of the entropy, suggesting that all the improved behavior is associated to an increased interlayer distance.

4. $\text{Na}_{0.62}\text{Mn}_{0.67}\text{Ni}_{0.21}\text{Cu}_{0.05}\text{Mg}_{0.016}\text{Mg}_{0.016}\text{Zn}_{0.016}\text{Sn}_{0.016}\text{Y}_{0.016}\text{O}_2$ and $\text{Na}_{0.62}\text{Mn}_{0.67}\text{Ni}_{0.21}\text{Cu}_{0.05}\text{Mg}_{0.016}\text{Mg}_{0.016}\text{Zn}_{0.016}\text{Sn}_{0.016}\text{Zr}_{0.016}\text{O}_2$ with 9 cations and higher entropy were prepared. What is the interlayer distance (or c lattice parameter) of these compounds?

5. Eq 1 in the paper is not sufficiently described. What are the conditions under which this equation is valid? Ref 28, cited in the text is related to "High-Entropy Liquids of Two Low-Melting Perylenes: A New Strategy for Liquid Chromophores". Does the solid-state condition change or alter the conditions in which the equation can be applied? More details are required from the authors to enable a full comprehension of the thermodynamic study to all readers. The info provided in the Supp info repeats the content given in the main text.

6. Authors claim that they observed possible participation of lattice O_2^- in the electrochemical process through XPS analysis. The new O component is observed in charge and disappears in discharge for both samples investigated. If the reaction is reversible, why in the CV plots the redox peak at 4.29V is not reversible for all samples? The peak at 3.9 in reduction is associated to $\text{Cu}^{2+}/\text{Cu}^{3+}$ and indeed is not there for NM cathode. More convincing explanation and results are required to confirm anionic redox processes. Does anionic redox affect phase stability or structural evolution if the materials? This is not highlighted in the in situ XRD patterns, where the materials keeps a P2 type structure upon charge/discharge (beside the small variation in of the unit cell volume).

7. P2 type cathodes sodium deficient phases, which generally leads to first charge capacities lower than the consequent discharge capacities. Fig S4 reports the voltage curves of the NM and CuMgTu-571 materials. From the figure, it is difficult to see the first cycle. Authors should specify the capacities achieved over the first cycle for both materials? From the normalized plots in fig 4 it appears that charge and discharge capacity are the same.

8. Authors claim that the kinetics of the anionic redox reaction are greatly improved by entropy. Can it be excluded that this is only an effect of the conductivity of the desodiated samples? Desodiated oxides are known to have lower conductivity (and generally increased impedance behavior, when compared to the sodiated phases). How can this be excluded and how can it be confirmed that the improvement is merely linked to the increased entropy? Authors should comment and discuss this in more detail.

Some other comments:

- In the introduction authors claim that "However, compared with Li^+ , the insertion/extraction of Na^+ in the electrodes shows poor reversibility and sluggish kinetics owing to its larger ionic radius (1.02 \AA for Na^+ vs. 0.76 \AA for Li^+)". This is not completely correct. Several studies have reported improved sodium-ion diffusion and kinetic when compared to lithium analogues (e.g. Aurbach et al., ACS Appl. Mater. Interfaces 2016, 8, 1867–1875). Ceder and co-workers (Ong, S. P.; Chevrier, V. L.; Hautier, G.; Jain, A.; Moore, C.; Kim, S.; Ma, X. H.; Ceder, G. Voltage, Stability and Diffusion Barrier Differences between Sodium-ion and Lithium-ion Intercalation Materials Energy Environ. Sci. 2011, 4, 3680–3688) showed that for NaCoO_2 , the diffusion barriers for sodium are lower than those for

lithium ions diffusion. Komaba showed that the alkali–oxygen bonds are longer in sodiated transition metal oxides, resulting in a weaker electrostatic interaction (Komaba, S.; Takei, C.; Nakayama, T.; Ogata, A.; Yabuuchi, N. *Electrochemical Intercalation Activity of Layered NaCrO₂ vs. LiCrO₂* *Electrochem. Commun.* 2010, 12, 355– 358). Most of these examples are on O3 type cathodes, if P2 type cathodes needs to be discussed, authors should review literature and rephrase the sentence.

- Page 19: A constant charge/discharge current of 20 mA g⁻¹ was applied by using a MACCOR system between 0.02 V and 2.0 V. This is not the correct voltage range.
- charge/discharge current of 20 mA g⁻¹ (what C rate does this correspond to, and which is the theoretical capacity of all the investigated materials? Please add info.
- Add amount electrolyte used in each electrochemical test.
- High purity Na-metal. What is the source? Is that a metal foil or have authors produced a metal foil from a precursor (e.g. metal ingots covered by oxide, or metal cubes under mineral oil?) Please add info.

Considering all the comments above, I suggest this paper to be submitted to an alternative journal only after careful evaluation of the comment above. A major reconsideration of the crucial aspects and scientific outcomes of this manuscript is strongly suggested.

Reviewer #2 (Remarks to the Author):

This manuscript aims to investigate the effect of high entropy that was induced in the materials through multiple cations doping, on the preferential crystal growth in layered oxides, which in turn could affect the electrochemical performance of these oxides used as cathode in Na cells. This topic is very recent, and more work should be done to ascertain the properties and exploit the applications of high entropy materials. Indeed, I have found this work interesting to read and well conducted. In my opinion it can be published in *Nat. Comm.* after few corrections. There's a couple of things I would ask the author to amend:

1) On pg. 8, lines 172-175 there is a comment concerning the effect of entropy on the growth rate of different crystal facets, that to the best of the authors' knowledge have not been exploited yet. I'm not against the consideration itself; it looks reasonable to me to relate those properties, but surface energies and thermodynamics of crystal growth have long been studied so I'm not sure it would be right to say "have not been exploited" even with a premise. This sentence rose my curiosity and I did a quick search. I can suggest two references as a starting point that might give some clue, i) R. Tran et. al, *Surface energies of elemental crystals*, *Scientific Data*, DOI: 10.1038/sdata.2016.80; ii) Talat S. Rahman, *Surface Thermodynamics and Vibrational Entropy*, *Springer Handbook of Surface Science*. Therefore, I am asking the authors, if possible, to find another way to express their uncertainty on studies relating entropy and growth rate, because I find that statement too much surprising considering how old is the topic for crystallographers and chemical physicists. Maybe readers could draw their own conclusions just saying "these results indicate that (configurational) entropy variation may reasonably alter the surface energy of crystal planes influencing the growth rate of different facets, which would need additional investigations beyond the primary scope of this work" or similar, in a way that the authors consider appropriate.

2) In this work I've found really a lot of information including the study of the high temperature structural changes by using in situ XRD. All considered, I think that there is no need to discuss samples such as Na_{0.62}Mn_{0.67}Ni_{0.21}Cu_{0.05}Mg_{0.016}Mg_{0.016}Zn_{0.016}Sn_{0.016}Y_{0.016}O₂ and Na_{0.62}Mn_{0.67}Ni_{0.21}Cu_{0.05}Mg_{0.016}Mg_{0.016}Zn_{0.016}Sn_{0.016}Zr_{0.016}O₂. This is kind of extra information, not really useful and not really interesting because the authors have explained already well their points with the CuMgTi-5ab samples. I would remove those samples and the relative figures and discussions from the main article and the supplementary information to make the reading lighter and fluid.

3) The cycling performance shows good results in terms of capacity for Na cells, but the coulombic

efficiency has not been described.

4) I think ref 22 has been given a wrong article title

5) I don't know if the session "Discussion" included the Conclusion since I can't find this latter.

6) I suggest including standard deviation at least for lattice parameters, cell V , obtained by Rietveld refinement.

A Point-by-point response to reviewers' comments

For Reviewer# 1

The manuscript reports on a novel strategy to develop ultrafast and ultra-stable P2-type layered cathodes by simultaneously and systematically regulating their configurational entropy and ion-diffusion tunnels. In situ synchrotron X-ray diffraction combined with electrochemical kinetic tests and microstructural characterization collectively revealed that the entropy-tuned $\text{Na}_{0.62}\text{Mn}_{0.67}\text{Ni}_{0.23}\text{Cu}_{0.05}\text{Mg}_{0.07}\text{Ti}_{0.01}\text{O}_2$ (CuMgTi-571) cathode possesses more {010} active facet, better structural stability, and thermal stability as well as 10-times faster anionic redox kinetics than that of $\text{Na}_{0.62}\text{Mn}_{0.67}\text{Ni}_{0.37}\text{O}_2$. Authors claim that their entropy and crystal facet modulation open up strategies to obtain high power and long-life sodium batteries. In their work they “have identified the hidden correlation between entropy and active crystal facet, cationic/anionic kinetics as well as electrochemical performance of P2-type cathodes”.

The overall study is well presented. However, the manuscript lacks in novelty and while the main outcome might be presented as highly innovative the main results confirm studies and observations already seen in the field. For this main reason it is suggested to submit the manuscript to a different journal. Some comments and questions highlighting these points are of the reasons are explained below:

General response: We thank the reviewer for the valuable comments to help us improve the quality of our manuscript. We are regret that we do not present you a clear information about the novelty of our work. We have provided a point-by-point response to your comments, as shown below:

Comment 1: High entropy cathode materials for SIBs have been already proposed by Hu et al. (Zhao, C. et al. High-Entropy Layered Oxide Cathodes for Sodium-Ion Batteries. *Angew. Chem. Int. Ed.* 59, 264-269 (2020)). More generally, high entropy oxides (HEOs), represented by the multi-element metallic systems that can crystallize in a single phase, is a known strategy. In this work, the novel aspect could be identified

with the fact that authors combine high entropy materials with active crystal facets. However, also this strategy has already been proposed. The same authors of this manuscript have reported on the same strategy on O3 type lithium compounds (ref 23 and 24 of this manuscript). The novelty is not very clear, beside the fact that this time the strategy is being applied to Na-based P2-type materials. In addition, considering how the percentage of {010} facets have been calculated (see page 7 and Fig. 3), it appears that the main difference among the investigated samples is associated to an increase of the interlayer distance ($d=11,16 \text{ \AA}$ vs 11.18 \AA for NM and CuMgTi-533, CuMgTi-552, CuMgTi-571 respectively), which certainly is beneficial for an improved kinetic of sodiation and de-sodiation enhancing rate capability. Overall, this phenomenon has already been comprehensively investigated and discussed in literature. The data are certainly well presented and represent a good piece of work, which however can be classified as incremental and does not represent an innovative breakthrough in the field. As such, publication in Nature Comm is not recommended.

Response 1: Thanks for your comments. To allow the reviewers to better understand the innovation of our work, the essential differences between our work and the other works are clarified as shown below:

(1) Comparison with sodium layered oxides with high entropy or increased {010} facets

Although Prof. Hu's group (Angew. Chem. Int. Ed. 2020, 59, 264-269) has reported the concept of high-entropy layered oxides cathodes for sodium-ion batteries, their system was based on O3 type sodium cathode at **low voltage region (2.0-3.9V)**, leading to relatively low specific capacity of $\sim 90 \text{ mAh g}^{-1}$ at 0.5 C. Meanwhile, the reported sodium layered oxide cathodes with increased {010} facets (ACS Appl. Mater. Interfaces 2019, 11, 30819-30827; Chemical Engineering Journal 2020, 382, 122978; Adv. Mater. 2018, 30, 1803765) were also evaluated within **low-voltage region (2.0-4.0 V)** only. In sharp contrast, we are working on sodium layered cathode towards **high-voltage operation (2.0-4.3 V)** to significant increase the specific capacity.

High-voltage operation is an effective way to increase the specific capacity of battery materials. However, it is well-known in battery community that high-voltage

operation often leads to undesired structural degradation and thus universal cycling instability. To date, despite numerous attempts through various strategies, the stabilization of layered cathodes at high voltage remains a formidable challenge. To the best of our knowledge, the correlation between entropy and high voltage cycling stability of layered cathodes remain elusive. The findings we gained in the present work has exactly filled this knowledge gap. Our results have clearly shown that the reversibility and kinetics of anionic redox at high-voltage can be greatly improved, and the irreversible P2→O2 phase transition during high-voltage charge can be also restrained through entropy tailoring and crystal-facet modulation. As a result, our optimal material exhibits extraordinary long-term cycling performance (87% capacity retention after 500 cycles at 1 C and 75.4% capacity retention after 2000 cycles at 10 C), outperforming many previous report results (Please see Table R1). In addition, the significant differences between our work and the aforementioned works are summarized in Table R2.

Table R1 Comparison of the cycling performance of various sodium-based P2 layered cathodes with CuMgTi-571 at same cut-off voltage.

Electrode	Voltage range (V)	Current density	Cycling performance	Ref.
CuMgTi-571	2.0-4.3	0.1 C (12 mA g ⁻¹)	89.6% (132.9 mAh g ⁻¹) after 100 cycles	This work
CuMgTi-571	2.0-4.3	1.0 C (12 mA g ⁻¹)	87% (132.9 mAh g ⁻¹) after 500 cycles	This work
P2-type Na _{2/3} Ni _{1/3} Mn _{2/3} O ₂	2.0-4.3	170 mA g ⁻¹	59.93% (53.9 mAh g ⁻¹) after 100 cycles	1
P2-type Na _{0.67} Ni _{0.17} Ti _{0.16} Mn _{0.67} O ₂	1.5-4.3	1.0 C	67% (113.9 mAh g ⁻¹) after 100 cycles.	2
P2- rhenanite-coated Na _{2/3} [Ni _{1/3} Mn _{2/3}]O ₂	2.5-4.3	40 mA g ⁻¹	74% (111 mAh g ⁻¹) after 200 cycles	3
P2-Na _{0.55} [Ni _{0.1} Fe _{0.1} Mn _{0.8}]O ₂	1.5-4.3	60 mA g ⁻¹	75% (102.9 mAh g ⁻¹) after 100 cycles	4
Na _{0.67} [Mn _{0.67} Ni _{0.21} Li _{0.06} Zn _{0.06}]O ₂	2.0-4.3	173 mA g ⁻¹	75% (137.3 mAh g ⁻¹) after 500 cycles	5
Co gradient P2- Na _{2/3} [Ni _{1/3} Mn _{2/3}]O ₂	2.0-4.3	16 mA g ⁻¹	77.4% (127.7 mAh g ⁻¹) after 100cycles	6

$\text{P2-Na}_{0.67}\text{Ni}_{0.33}\text{Mn}_{0.47}\text{Ti}_{0.2}\text{O}_2$	2.5-4.3	173 mA g ⁻¹	78.6% (88.1 mAh g ⁻¹) after 200 cycles	7
$\text{P2-Na}_{2/3}[\text{Ni}_{0.3}\text{Co}_{0.1}\text{Mn}_{0.6}]\text{O}_2$	2.0-4.3	15 mA g ⁻¹	79.2% (127.9 mAh g ⁻¹) after 50 cycles	8
$\text{P2-Na}_{0.66}\text{Co}_{0.22}\text{Mn}_{0.44}\text{Ti}_{0.34}\text{O}_2$	1.5-4.3	35.4 mA g ⁻¹	79.4% (104 mAh g ⁻¹) after 100 cycles	9

Table R2 The significant differences between our work and sodium layered cathodes with high-entropy or increased crystal facets.

Reference	Strategy	Material type	Voltage range (V)	Capacity/Capacity retention/Cycle/C-rate
Our work	Entropy control and crystal-facet tuning	P2-type	2.0-4.3	132.9 mAh g ⁻¹ /89.6%/100 cycles/0.1 C, 103.2 mAh g ⁻¹ /87%/500 cycles/1 C
Hu's work (Angew. Chem. Int. Ed. 59, 264-269 (2020))	Entropy tuning	O3-type	2.0-3.9	90 mAh g ⁻¹ /85.7%/200 cycles /0.5 C
ACS Appl. Mater. Interfaces 2019, 11, 30819-30827	Crystal-facet tuning	P2-type	2.0-4.0	70 mAh g ⁻¹ /86%/1200 cycles /10 C
Chemical Engineering Journal 2020, 382, 122978	Crystal-facet tuning	O3-type	2.0-4.0	110 mAh g ⁻¹ /77.3%/100 cycles/ 0.15 C
Adv. Mater. 2018, 30, 1803765	Crystal-facet tuning	O3-type	2.0-4.0	100 mAh g ⁻¹ /86.9%/600 cycles/5.0 C

(2) Comparison with our previous work about crystal facets modulation on O3 lithium compounds (ref 23 and 24 of this manuscript, J. Mater. Chem. A. 2013, 1, 3860-3864 and Adv. Mater. 2010, 22, 4364-4367)

Although layered cathode materials with increased exposure of {010} active facets were synthesized in our works, we would like to point out that the impetus of forming high percentage {010} facets is essentially different.

Table R3 The differences between our present work and previous works in four aspects.

Reference	Composition	Configurational entropy	Key factor affecting surface energy	Impetus for growth of high percentage {010} facets
This work	$\text{Na}_{0.62}\text{Mn}_{0.67}\text{Ni}_{0.23}\text{Cu}_0$ $.05\text{Mg}_{0.09-2y}\text{Ti}_y\text{O}_2$	1.26-1.28R	Configurational entropy	Configurational entropy
J. Mater. Chem. A. 1, 3860-3864 (2013)	$\text{LiNi}_{1/3}\text{Co}_{1/3}\text{Mn}_{1/3}\text{O}_2$	1.09R	Temperature	Temperature
Adv. Mater. 22, 4364-4367 (2010)	$\text{Li}(\text{Li}_{0.17}\text{Ni}_{0.25}\text{Mn}_{0.58})\text{O}_2$	0.95R	Temperature	Temperature

Based on the calculation formula of surface energy $\gamma(T)=\gamma_0 - (T/A)(S_{\text{vib}} + S_{\text{conf}})$ (Where $\gamma(T)$ and γ_0 are surface energy at any temperature and 0 K, respectively. T is temperature, A is Helmholtz free energy, S_{vib} and S_{conf} are system's vibrational and configurational entropy, respectively.), it can be found that the surface energy of system with simple composition and low entropy is mainly influenced by temperature, while the surface energy of polycomponent system with high entropy is mainly influenced by configurational entropy S_{conf} . [see ref 10]

The P2-type layered cathode materials in our current work contain 6 or 9 elements, which are high entropy system. The surface energy of these P2-type entropy-tuned materials is mainly affected by configurational entropy S_{conf} at a given temperature. Thus, high configurational entropy is the impetus for the growth of high percentage {010} active facets. In contrast, the compositions of our O3 lithium compounds ($\text{LiNi}_{1/3}\text{Co}_{1/3}\text{Mn}_{1/3}\text{O}_2$ and $\text{Li}(\text{Li}_{0.17}\text{Ni}_{0.25}\text{Mn}_{0.58})\text{O}_2$) are simple, which only contain 3 and 4 elements. The surface energy of these compounds is mainly affected by temperature, and temperature is the driving force for the growth of {010} active facets. Their differences are clearly shown in Table R3.

In brief, we have two critical findings that have not been reported in the existing literature. These findings are valuable for design of high-voltage cathodes with superior cycling stability and rate capability.

- 1) **Structure design:** The surface energy and active facets of layered oxides materials can be tuned by changing their configurational entropy, which provides new material design principles for high-performance cathode materials.
- 2) **Performance tuning:** The reversibility of phase transition and durability/kinetics of anionic redox in layered oxides cathode materials during high-voltage operation can be greatly improved by entropy control and crystal facet modulation, significantly improving the energy density and cycle life of layered cathode materials.

[1] Zheng, X. et al. New insights into understanding the exceptional electrochemical performance of P2-Type manganese-based layered oxide cathode for sodium ion batteries. *Energy Stor. Mater.* **15**, 257-265 (2018).

[2] Pahari, D. et al. On controlling the P2-O2 phase transition by optimal Ti-substitution on Nisite in P2-type $\text{Na}_{0.67}\text{Ni}_{0.33}\text{Mn}_{0.67}\text{O}_2$ (NNMO) cathode for Na-ion batteries. *J. Power Sources.* **455**, 227957 (2020).

[3] Jo, C. et al. Bioinspired surface layer for the cathode material of high-energy-density sodium-ion batteries. *Adv. Energy Mater.* **8**, 1702942 (2018).

[4] Hwang, J. et al. A new P2-Type layered oxide cathode with extremely high energy density for sodium-ion batteries. *Adv. Energy Mater.* **9**, 1803346 (2019).

[5] Li, W. et al. Exploring the stability effect of the Co-substituted P2- $\text{Na}_{0.67}[\text{Mn}_{0.67}\text{Ni}_{0.33}]\text{O}_2$ cathode for liquid- and solid-state sodium ion batteries. *ACS Appl. Mater. Interfaces.* **12**, 41477-41484 (2020).

[6] Hou, P. et al. Mitigating the P2–O2 phase transition of high-voltage P2- $\text{Na}_{2/3}[\text{Ni}_{1/3}\text{Mn}_{2/3}]\text{O}_2$ cathode by cobalt gradient substitution for high-rate sodium-ion batteries. *J. Mater. Chem. A*, **7**, 4705-4713 (2019).

[7] Tang, K. et al. Electrochemical performance and structural stability of air-stable $\text{Na}_{0.67}\text{Ni}_{0.33}\text{Mn}_{0.67-x}\text{Ti}_x\text{O}_2$ cathode materials for high-performance sodium ion batteries. *Chem. Eng. J.* **399**, 125725 (2020).

[8] Hou, P. et al. A high energy-density P2- $\text{Na}_{2/3}[\text{Ni}_{0.3}\text{Co}_{0.1}\text{Mn}_{0.6}]\text{O}_2$ cathode with mitigated P2–O2 transition for sodium-ion battery. *Nanoscale*, **11**, 2787-2794 (2019).

[9] Wang, Q. et al. Utilizing $\text{Co}^{2+}/\text{Co}^{3+}$ redox couple in P2-Layered $\text{Na}_{0.66}\text{Co}_{0.22}\text{Mn}_{0.44}\text{Ti}_{0.34}\text{O}_2$ cathode for sodium-ion batteries. *Adv. Sci.* **4**, 1700219 (2017).

[10] Rocca, M., Rahman, S. & Vattuone, L. *Springer Handbook of Surface Science*. (Springer, 2020).

Comment 2: Authors claim that: “their results indicate that entropy variation may alter the surface energy of crystal planes, and further influences the growth rate of different facets. To the best of our knowledge, the effect of the entropy on the growth rate of crystal planes have not been exploited and required more comprehensive study.” The high entropy and the active crystal facets are a consequence of the introduction in the materials of multiple metal ions with larger ionic radii [larger ionic sizes of Cu^{2+} (0.73 Å) and Mg^{2+} (0.72 Å) compared to Mn^{4+} (0.53 Å) and Ni^{2+} (0.69 Å)]. This induces not only a higher entropy but also a larger interlayer distance into the layered structure, suggesting that the main effect on the improved rate capability is associated only to a larger interlayer distance. As a matter of fact, authors claim that “increasing configuration entropy may be beneficial for improving the growth rate of layered oxide cathodes along the [001] direction and promote the formation of layered cathode with more {010} facets”. What would have happened if the increased entropy would have been achieved by using different metals, for instance with smaller ionic radii? To evaluate this, author should be able to prepare sample with high entropy by using smaller ionic radii substituents. The comparison will elucidate the differences. Would then the materials still have active crystal facets?

Response 2: We thank but respectively cannot agree with the reviewer on these comments.

To clarify the effect of ionic radii of substitution elements on interlayer distance and active facet of P2-type layered cathodes, a high entropy (1.21R) sample $\text{Na}_{0.62}\text{Mn}_{0.67}\text{Ni}_{0.23}\text{Co}_{0.033}\text{Al}_{0.033}\text{Ti}_{0.02}\text{O}_2$ (CoAlTi-332) consisting of small-sized cations (Co^{3+} :0.61Å, Al^{3+} :0.53 Å, Ti^{4+} :0.60 Å) has been prepared. The XRD pattern and Rietveld refinement of CoAlTi-332 are displayed in Fig. R1a. It is clear that all the diffraction peaks could be assigned to the typical P2-typed structure. Table R4 compares the lattice parameter c and interlayer distance (d_{0-0}) of CoAlTi-332 and

CuMgTi-571 deduced from the Rietveld. The results show that **introducing cations with smaller ionic radii into P2-typed structure can lead to increased interlayer spacing**. In addition, SEM image of CoAlTi-332 in Fig. R1b shows that CoAlTi-332 consists of microbricks with thickness of 0.7-1.1 μm , which is similar to that of CuMgTi-571, indicating that CoAlTi-332 consisting of small-sized cations could also has high active facets. Moreover, reference [1] also suggested that **lattice parameter c doesn't always increase with increased substitution amount of large-sized cation**. The lattice parameter c of $\text{Na}_{0.72}\text{Li}_{0.24}\text{Ti}_x\text{Mn}_{0.76-x}\text{O}_2$ expands when Ti substitution amount increases from $x=0.05$ to $x=0.1$, while contracts when Ti substitution amount increases from $x=0.1$ to $x=0.2$.

The lattice parameters c (interlayer distance) of different P2-type Mn-based materials in the literatures were also compared in Table R5. It can be clearly seen that **the lattice parameter c doesn't linearly increase with increased radius of substituted cation, either**. In addition, it can also be found that configurational entropy has no linear relation with lattice parameter c . The above results demonstrate that active facets, interlayer spacing, and entropy are not directly proportional to the ionic radii of substituent.

Engineering Na^+ -layer spacings could indeed improve the performance of P2 cathode (e.g., from 3.6 \AA to 5.8 \AA in Nature Commun. 2021, 12, 4903). However, in the present work, the maximum changes of interlayer distance (d) are only 0.11 \AA . Such a small variation of interlayer distance is not sufficient to account for the great improvement in the rate capability. In contrast, the configurational entropy of CuMgTi-533, CuMgTi-552, CuMgTi-571 and NaMnO_2 were 1.26R, 1.27R, 1.28R and 0.93R, respectively. The relatively larger entropy difference (e.g., 0.35R between CuMgTi-571 and NaMnO_2) should play a dominant role in the observed improvement on rate capability.

Fig. R1 (a) XRD pattern, Rietveld refinement plot and (b) SEM image of $\text{Na}_{0.62}\text{Mn}_{0.67}\text{Ni}_{0.23}\text{Co}_{0.033}\text{Al}_{0.033}\text{Ti}_{0.02}\text{O}_2$ (CoAlTi-332).

Table R4 The differences between CuMgTi-571 and CoAlTi-332.

Materials with different compositions	Ionic radii	Lattice parameter ©	Interlayer distance
$\text{Na}_{0.62}\text{Mn}_{0.67}\text{Ni}_{0.23}\text{Cu}_{0.05}\text{Mg}_{0.07}\text{Ti}_{0.01}\text{O}_2$	Cu^{2+} (0.73 Å) Mg^{2+} (0.72 Å)	11.18194 Å	4.0948 Å
$\text{Na}_{0.62}\text{Mn}_{0.67}\text{Ni}_{0.23}\text{Co}_{0.033}\text{Al}_{0.033}\text{Ti}_{0.02}\text{O}_2$	Co^{3+} (0.61 Å) Al^{3+} (0.53 Å)	11.22436 Å	4.1103 Å

Table R5 shows the comparison of c value and configurational entropy between different P2-type Mn-based materials.

Composition	Configurational entropy	Ionic radii of substituted cation	Lattice parameter ©(c)	Ref.
$\text{Na}_{2/3}\text{Fe}_{0.2}\text{Mn}_{0.8}\text{O}_2$	0.77R	Fe^{3+} (0.55 Å)	11.2042 Å	2
$\text{Na}_{2/3}\text{Ni}_{1/3}\text{Mn}_{2/3}\text{O}_2$	0.91R	Ni^{2+} (0.69 Å)	11.155 Å	3
$\text{Na}_{0.66}\text{Mn}_{0.9}\text{Mg}_{0.1}\text{O}_2$	0.60R	Mg^{2+} (0.72 Å)	11.324 Å	4
$\text{Na}_{2/3}\text{Ni}_{1/12}\text{Cu}_{1/4}\text{Mn}_{2/3}\text{O}_2$	1.09R	Ni^{2+} (0.69 Å) Cu^{2+} (0.73 Å)	11.15 Å	5
$\text{Na}_{2/3}[\text{Zn}_{0.3}\text{Mn}_{0.7}]\text{O}_2$	0.88R	Zn^{2+} (0.74 Å)	11.1465 Å	6

[1] Li, C. et al. Unraveling the critical role of Ti substitution in P2- $\text{Na}_x\text{Li}_y\text{Mn}_{1-y}\text{O}_2$ cathode for highly reversible oxygen redox chemistry. *Chem. Mater.* **32**, 1054-1063 (2020).

- [2] Dose, W. et al. Structure-electrochemical evolution of a Mn-rich P2 $\text{Na}_{2/3}\text{Fe}_{0.2}\text{Mn}_{0.8}\text{O}_2$ Na-ion battery cathode. *Chem. Mater.* **29**, 7416-7423 (2017).
- [3] Zhao, C. et al. Revealing high Na-content P2-type layered oxides as advanced sodium-ion cathodes. *J. Am. Chem. Soc.* **142**, 5742-5750 (2020).
- [4] Kaliyappan, K. et al. Constructing safe and durable high-voltage P2 layered cathodes for sodium ion batteries enabled by molecular layer deposition of alucone. *Adv. Funct. Mater.* **30**, 1910251 (2020).
- [5] Zheng, L., Li, J. & Obrovac, M. Crystal structures and electrochemical performance of air-stable $\text{Na}_{2/3}\text{Ni}_{1/3-x}\text{Cu}_x\text{Mn}_{2/3}\text{O}_2$ in sodium cells. *Chem. Mater.* **29**, 1623-1631 (2017).
- [6] Konarov, A. et al. High-voltage oxygen-redox-based cathode for rechargeable sodium-ion batteries. *Adv. Energy Mater.* **10**, 2001111 (2020).

Comment 3: Table 6 in Supp Info provides calculated entropy value for literature data. It would be interesting to see if the d spacing of the materials reported present the same trend of the entropy, suggesting that all the improved behavior is associated to an increased interlayer distance.

Response 3:

Table R6 Comparison of c value and electrochemical performance of this work with other reported P2 layered cathodes.

Cathode materials	Entropy and lattice parameter c	Voltage range [V]	Current density	Cycle number	Capacity retentions	Rate properties (mAh g ⁻¹)	Line color	Ref.
$\text{Na}_{0.42}\text{Mn}_{0.87}\text{Ni}_{0.23}\text{Cu}_{0.03}\text{Mg}_{0.07}\text{Ti}_{0.01}\text{O}_2$	$S_{\text{conf}}=1.28\text{R}$ $C=11.18 \text{ \AA}$	2.0-4.3	10 C	3000 1000 500	68% 87.4% 95.4%	78.6/10C	—	This work
$\text{Na}_{4.552}\text{Li}_{4.54}\text{Ni}_{16.54}\text{Mn}_{34.54}\text{O}_{72}$	$S_{\text{conf}}=1.00\text{R}$ $C=11.07 \text{ \AA}$	2.0-4.0	3 C	3000	68%	60/10C	—	ref. 38
$\text{Na}_{0.7}\text{Mn}_{0.6}\text{Ni}_{0.2}\text{Mg}_{0.2}\text{O}_2$	$S_{\text{conf}}=1.20\text{R}$ $C=11.15 \text{ \AA}$	2.5-4.2	1 C	1000	79%	70/10C	—	ref.4
$\text{Na}_{2.3}\text{Ni}_{1.6}\text{Mn}_{2.3}\text{Cu}_{1.9}\text{Mg}_{1.18}\text{O}_2$	$S_{\text{conf}}=1.24\text{R}$ $C=11.19 \text{ \AA}$	2.5-4.15	5 C	500	81.4%	78/10C	—	ref.39
$\text{Na}_{0.76}\text{Cu}_{0.22}\text{Fe}_{0.30}\text{Mn}_{0.48}\text{O}_2$	$S_{\text{conf}}=1.26\text{R}$ $C=11.21 \text{ \AA}$	2.0-4.0	2 C	300	79%	64.9/10C	—	ref.40
$\text{Na}_{2.3}\text{Mn}_{0.72}\text{Cu}_{0.22}\text{Mg}_{0.06}\text{O}_2$	$S_{\text{conf}}=1.01\text{R}$ $C=11.19 \text{ \AA}$	2.0-4.5	1 C	100	87.9%	70.3/10C	—	ref.41
$\text{Na}_{0.67}\text{Mn}_{0.71}\text{Cu}_{0.02}\text{Mg}_{0.02}\text{Ni}_{0.25}\text{O}_2$	$S_{\text{conf}}=1.01\text{R}$ $C=11.18 \text{ \AA}$	1.5-4.5	1 C	100	81.9%	72/10C	—	ref.42

Thanks for this comment. The literatures in Supplementary Table 6 all give the c lattice parameter of the materials, but unfortunately doesn't provide the value of d spacing. Thus, we can only summarize lattice parameters c of the literatures and list in Table R6. It can be seen that the lattice parameters c of the materials doesn't show the

same trend of the entropy and electrochemical performance. And as explained in the response 2, in comparison with the entropy, the impact of interlayer spacing does not play a dominant role in the electrochemical performance of our layered oxides cathodes.

Comment 4: $\text{Na}_{0.62}\text{Mn}_{0.67}\text{Ni}_{0.21}\text{Cu}_{0.05}\text{Mg}_{0.016}\text{Mg}_{0.016}\text{Zn}_{0.016}\text{Sn}_{0.016}\text{Y}_{0.016}\text{O}_2$ (1.31R) and $\text{Na}_{0.62}\text{Mn}_{0.67}\text{Ni}_{0.21}\text{Cu}_{0.05}\text{Mg}_{0.016}\text{Mg}_{0.016}\text{Zn}_{0.016}\text{Sn}_{0.016}\text{Zr}_{0.016}\text{O}_2$ (1.29R) with 9 cations and higher entropy were prepared. What is the interlayer distance (or c lattice parameter) of these compounds?

Response 4: The Rietveld refinement analysis of $\text{Na}_{0.62}\text{Mn}_{0.67}\text{Ni}_{0.21}\text{Cu}_{0.05}\text{Mg}_{0.016}\text{Mg}_{0.016}\text{Zn}_{0.016}\text{Sn}_{0.016}\text{Y}_{0.016}\text{O}_2$ (HEO-Y) and $\text{Na}_{0.62}\text{Mn}_{0.67}\text{Ni}_{0.21}\text{Cu}_{0.05}\text{Mg}_{0.016}\text{Mg}_{0.016}\text{Zn}_{0.016}\text{Sn}_{0.016}\text{Zr}_{0.016}\text{O}_2$ (HEO-Zr) are presented in Fig. R2a and R2b. The interlayer distances (d_{o-o}) for HEO-Y and HEO-Zr obtained from the refinement are 4.0949 Å and 4.0933 Å, respectively. The d_{o-o} of HEO-Y is close to that of HEO-Zr, although the radius of Y^{3+} (0.9 Å) is much larger than that of Zr^{4+} (0.72 Å), once again confirmed that the interlayer spacing is not linearly increased with size of substitution cations.

Fig. R2 XRD patterns and Rietveld refinement plots of $\text{Na}_{0.62}\text{Mn}_{0.67}\text{Ni}_{0.21}\text{Cu}_{0.05}\text{Mg}_{0.016}\text{Mg}_{0.016}\text{Zn}_{0.016}\text{Sn}_{0.016}\text{Y}_{0.016}\text{O}_2$ (HEO-Y) and $\text{Na}_{0.62}\text{Mn}_{0.67}\text{Ni}_{0.21}\text{Cu}_{0.05}\text{Mg}_{0.016}\text{Mg}_{0.016}\text{Zn}_{0.016}\text{Sn}_{0.016}\text{Zr}_{0.016}\text{O}_2$ (HEO-Zr).

Comment 5: Eq 1 in the paper is not sufficiently described. What are the conditions under which this equation is valid? Ref 28, cited in the text is related to “High-Entropy Liquids of Two Low-Melting Perylenes: A New Strategy for Liquid Chromophores”.

Does the solid-state condition change or alter the conditions in which the equation can be applied? More details are required from the authors to enable a full comprehension of the thermodynamic study to all readers. The info provided in the Supp info repeats the content given in the main text.

Response 5: Thanks for the suggestion. Eq 1 in the manuscript can be used in ideal gas mixture, ideal or regular solution, and solid compound with single-phase crystal structure.[1-4] The using condition of Eq 1 was added into the Supp info and the duplicate content in Supp info has been deleted.

$$S_{\text{config}} = -R \left[\left(\sum_{i=1}^N x_i \ln x_i \right)_{\text{cation-site}} + \left(\sum_{j=1}^M x_j \ln x_j \right)_{\text{anion-site}} \right] \quad (\text{Eq. 1})$$

[1] Kushwaha, K. et al. A record chromophore density in high-entropy liquids of two low-melting perylenes: a new strategy for liquid chromophores. *Adv. Sci.* **6**, 1801650 (2019).

[2] Abhishek Sarkar, A. et al. High-Entropy Oxides: Fundamental Aspects and Electrochemical Properties. *Adv. Mater.* **31**, 1806236 (2019).

[3] Abhishek Sarkar, A. et al. High entropy oxides for reversible energy storage. *Nat. Commun.* **9**, 3400 (2018).

[4] Santodonato, L. et al. Deviation from high-entropy configurations in the atomic distributions of a multi-principal-element alloy. *Nat. Commun.* **6**, 5964 (2015).

Comment 6: Authors claim that they observed possible participation of lattice O_2^- in the electrochemical process through XPS analysis. The new O component is observed in charge and disappears in discharge for both samples investigated. If the reaction is reversible, why in the CV plots the redox peak at 4.29V is not reversible for all samples? The peak at 3.9 in reduction is associated to $\text{Cu}^{2+}/\text{Cu}^{3+}$ and indeed is not there for NM cathode. More convincing explanation and results are required to confirm anionic redox processes. Does anionic redox affect phase stability or structural evolution of the materials? This is not highlighted in the in situ XRD patterns, where the materials keeps a P2 type structure upon charge/discharge (beside the small variation in of the unit cell volume).

Response 6: Thanks for these comments.

(1) The weak peak at around 3.81 V in reduction is associated to $\text{Cu}^{2+}/\text{Cu}^{3+}$, rather than 3.96 V as cited by the reviewer. The peak intensity is weak, which is due to the low content of Cu element. Anionic redox reaction ($\text{O}^{2-}/\text{O}_2^{n-}$) ($1 \leq n \leq 3$) are both reversible in the first CV curves of CuMgTi-571 and NaMNO_2 (Fig. R3). However, they show significant difference on the voltage hysteresis (e.g., the oxidation and reduction peak of anionic redox locates at different potentials). The hysteresis phenomenon is often observed in the cathode materials with anionic redox.[1-4] The peak potential differences (ΔE_p) of anionic redox reaction in NaMNO_2 is 0.59 V (4.30 V and 3.71 V for oxidation and reduction peak, respectively). In contrast, the (ΔE_p) of anionic redox reaction in CuMgTi-571 is decreased to 0.34 V (4.29 V and 3.95 V for oxidation and reduction peak, respectively). Therefore, they exhibit significant different stability of oxygen redox after extended cycles. As shown in the dQ/dV curves of CuMgTi-571 and NaMNO_2 after 50 cycles (Fig. R4), the anionic redox peaks at high voltage have been well maintained in the CuMgTi-571 in comparison with NaMNO_2 .

Fig. R3 First CV curves of $\text{Na}_{0.62}\text{Mn}_{0.67}\text{Ni}_{0.23}\text{Cu}_{0.05}\text{Mg}_{0.09-2y}\text{Ti}_y\text{O}_2$ and NaMNO_2 electrodes between 2 and 4.3 V at 0.2 mV s^{-1} scan rate.

Fig. R4 Differential capacity vs. voltage (dQ/dV) plots of CuMgTi-571 and NaMnO₂ in the 50th cycle at 0.1 C rate.

(2) XPS has been used to study the redox reactions on the O by Tarascon et al. and Goodenough et al.[6-9] In the present work, we therefore employed XPS to probe O evolution at different charge/discharge states. We are aware of the newly developed techniques to probe the oxygen redox, such as O K-edge mapping of resonant inelastic X-ray scattering by Dr. Wanli Yang at LBNL (Joule 2019, 3, 518-541). However, due to the pandemic, the access to this technique is very limited at this time. We hope the reviewer can understand this circumstance.

(3) Anionic redox usually results in generation of oxygen vacancies and migration of transition metal (TM) ions, which could cause phase transition and new phase formation. The drastic phase transition (P2-(OP4)-O2) at high voltage in P2-type Mn-based layered oxide has been widely observed in the literatures. In contrast, no sign of the formation of O2 or OP4 phase (or Z phase) are observed in the in situ HEXRD patterns of CuMgTi-571 during charge/discharge. Moreover, the in-situ HEXRD during heating in Fig. 7a and 7b (in manuscript) clearly showed that the on-set temperature for the phase transition of high-voltage charged CuMgTi-571 (after triggering anionic redox) during heating has been postponed to a higher temperature in comparison with that of charged NaMnO₂, indicating better structural stability.

- [1] Hwang, J., Kim, J., Yu, T. & Sun, Y. A new P2-type layered oxide cathode with extremely high energy density for sodium-ion batteries. *Adv. Energy Mater.* **9**, 1803346 (2019).
- [2] Kaliyappan, K. et al. An ion conductive polyimide encapsulation: new insight and significant performance enhancement of sodium based P2 layered cathodes. *Energy Stor. Mater.* **22**, 168-178 (2019).
- [3] Sun, C. et al. Construction of the $\text{Na}_{0.92}\text{Li}_{0.40}\text{Ni}_{0.73}\text{Mn}_{0.24}\text{Co}_{0.12}\text{O}_2$ sodium-ion cathode with balanced high-power/energy-densities. *Energy Stor. Mater.* **27**, 252-260 (2020).
- [4] Zheng, L., Li, J. & Obrovac, M. Crystal structures and electrochemical performance of air-stable $\text{Na}_{2/3}\text{Ni}_{1/3-x}\text{Cu}_x\text{Mn}_{2/3}\text{O}_2$ in sodium cells. *Chem. Mater.* **29**, 1623-1631 (2017).
- [5] Assat, G. et al. Fundamental interplay between anionic/cationic redox governing the kinetics and thermodynamics of lithium-rich cathodes. *Nat. Commun.* **8**, 2219 (2017).
- [6] Sathiya, M. et al. Origin of voltage decay in high-capacity layered oxide electrodes. *Nat. Mater.* **14**, 230-238 (2015).
- [7] Perez, A. et al. Strong oxygen participation in the redox governing the structural and electrochemical properties of Na-rich layered oxide Na_2IrO_3 . *Chem. Mater.* **28**, 8278-8288 (2016).
- [8] Du, K. et al. Exploring reversible oxidation of oxygen in a manganese oxide. *Energy Environ. Sci.* **9**, 2575-2577 (2016).
- [9] Pearce, P. et al. Evidence for anionic redox activity in a tridimensional-ordered Li-rich positive electrode $\beta\text{-Li}_2\text{IrO}_3$. *Nat. Mater.* **16**, 580-586 (2017).

Comment 7: P2 type cathodes sodium deficient phases, which generally leads to first charge capacities lower than the consequent discharge capacities. Fig S4 reports the voltage curves of the NM and CuMgTi-571 materials. From the figure, it is difficult to see the first cycle. Authors should specify the capacities achieved over the first cycle for both materials? From the normalized plots in fig 4 it appears that charge and discharge capacity are the same.

Response 7: We would like to clarify that the normalization of voltage-capacity curve is aiming to clearly display the voltage fade issue of two cathode materials with anionic redox, which clearly showed that CuMgTi-571 exhibit much less voltage fade. The charge/discharge curves of CuMgTi-571 and NaMnO_2 cathode were provided in Supplementary Fig. 6.

We have now compared the 1st charge/discharge curves of CuMgTi-571 and NaMnO_2 cathode (Fig. R5). As shown, CuMgTi-571 displays a smoother charge-

discharge curve than NaMNO₂, indicating suppressed phase transition. In addition, CuMgTi-571 delivers an initial charge capacity of 128.9 mAh g⁻¹ and discharge capacity of 148.2 mAh g⁻¹, respectively. Differently, NaMNO₂ shows an initial charge and discharge capacities of 162.8 and 150.5 mAh g⁻¹, respectively.

We have provided this information more clearly in the revised manuscript.

Fig. R5 First charge-discharge curves of CuMgTi-571 and NaMNO₂ at 0.1 C rate tested between 2 and 4.3 V.

Comment 8: Authors claim that the kinetics of the anionic redox reaction are greatly improved by entropy. Can it be excluded that this is only an effect of the conductivity of the desodiated samples? Desodiated oxides are known to have lower conductivity (and generally increased impedance behavior, when compared to the sodiated phases). How can this be excluded and how can it be confirmed that the improvement is merely linked to the increased entropy? Authors should comment and discuss this in more detail.

Response 8: We thank the reviewer for the comments on the effect of electronic conductivity on the electrochemical performance. To address your concern, in-situ EIS experiments were conducted on CuMgTi-571 and NaMNO₂ to examine the effect of the conductivity on anionic redox reaction at high voltage.

Fig. R6 show the Nyquist plots of the CuMgTi-571 and NaMNO₂ at different potentials during the first charge and discharge process. Upon Na⁺ insertion/extraction, both samples experience a series of processes in the bulk and at the surface, leading to

different suppressed semicircles in Nyquist plots of the first charge and discharge process. At high voltage region (4.0-4.3 V), the Nyquist plots of both samples exhibit two distinct parts including a depressed semicircle in the high-frequency region and an inclined line in the low-frequency zone. The high-frequency semicircle should be related to the charge-transfer resistance. By comparing the diameters of the semicircles at high voltage region (Table R7), the impedances of CuMgTi-571 and NaMNO₂ are similar, implying similar electronic conductivity of the two desodiated samples. Therefore, the significantly improved kinetics of the anionic redox in the CuMgTi-571 is mainly contributed from its unique structure with high active facets and high entropy, which can provide more stable migration tunnels and is helpful for accelerating Na⁺ diffusion.

The data and analysis of in-situ EIS have been added in the revised manuscript and Supplementary information.

Fig. R6 in-situ EIS spectra of (a,b) CuMgTi-571 and (c,d) NaMNO₂ at different potentials of the first charge and discharge process at 0.1 C.

Table R7 Diameters of semicircles in EIS spectra of CuMgTi-571 and NaMNO₂ at high voltage region (4.0-4.3 V).

High-voltage region	Diameter of semicircle in EIS spectra of CuMgTi-571	Diameter of semicircle in EIS spectra of NaMNO ₂
CC 4.0 V	132.6 Ω	131.8 Ω
CC 4.1 V	133.8 Ω	138.0 Ω
CC 4.2 V	86.5 Ω	89.5 Ω
CC 4.3 V	78.7 Ω	84.2 Ω
DC 4.2 V	73.9 Ω	87 Ω
DC 4.1 V	75.7 Ω	98.6 Ω
DC 4.0 V	135.6 Ω	162 Ω

Comment 9: In the introduction authors claim that “However, compared with Li⁺, the insertion/extraction of Na⁺ in the electrodes shows poor reversibility and sluggish kinetics owing to its larger ionic radius (1.02 Å for Na⁺ vs. 0.76 Å for Li⁺)”. This is not completely correct. Several studies have reported improved sodium-ion diffusion and kinetic when compared to lithium analogues (e.g. Aurbach et al., ACS Appl. Mater. Interfaces 2016, 8, 1867–1875). Ceder and co-workers (Ong, S. P.; Chevrier, V. L.; Hautier, G.; Jain, A.; Moore, C.; Kim, S.; Ma, X. H.; Ceder, G. Voltage, Stability and Diffusion Barrier Differences between Sodium-ion and Lithium-ion Intercalation Materials Energy Environ. Sci. 2011, 4, 3680– 3688) showed that for NaCoO₂, the diffusion barriers for sodium are lower than those for lithium ions diffusion. Komaba showed that the alkali–oxygen bonds are longer in sodiated transition metal oxides, resulting in a weaker electrostatic interaction (Komaba, S.; Takei, C.; Nakayama, T.; Ogata, A.; Yabuuchi, N. Electrochemical Intercalation Activity of Layered NaCrO₂ vs. LiCrO₂ Electrochem. Commun. 2010, 12, 355–358). Most of these examples are on O3 type cathodes, if P2 type cathodes needs to be discussed, authors should review literature and rephrase the sentence.

Response 9: We appreciate the reviewer for pointing out these references.

To investigate and compare the Li⁺ kinetics and Na⁺ kinetics in electrode materials, the diffusion coefficients of the common P2 Na-based layered oxides and Li-based layered oxides studied in the literatures are summarized and listed in Table R8. It can be seen that the Na⁺ diffusion coefficients in many P2 Na-based layered oxides are generally lower than 10⁻¹⁰ cm² s⁻¹, which are smaller than those of the Li-based layered oxides (mostly at magnitudes of ≈10⁻⁸–10⁻¹⁰ cm² s⁻¹). Thus, Na⁺ diffusion dynamics is in general slower than Li⁺ diffusion dynamics in P2 layered cathode materials, owing to the larger size of Na⁺.

In the revised manuscript, we have cited these references and revised the sentence as “However, compared with Li⁺, the insertion/extraction of Na⁺ in the reported P2 cathodes generally shows poor reversibility and sluggish kinetics owing to its larger ionic radius (1.02 Å for Na⁺ vs. 0.76 Å for Li⁺) (Supplementary Table 1).”

Table R8. Summary of diffusion coefficients of the Na-based layered oxides and Li-based layered oxides reported in the literature using GITT tests.

Compound	D (cm ² s ⁻¹)	Ref.
O3-LiNi _{0.8} Co _{0.1} Mn _{0.1} O ₂ @PANI-PVP	0.5×10 ⁻⁸ -3.5×10 ⁻⁸	1
O3-LiNi _{0.88} Co _{0.09} Al _{0.03} O ₂ -Te 1%	10 ⁻⁹	2
O3-LiNi _{0.6} Co _{0.2} Mn _{0.2} O ₂	10 ⁻⁹	3
O3-LiNi _{0.9} Co _{0.07} Al _{0.03} O ₂	10 ⁻⁹	4
O3-Li _{1.2} Mn _{0.54} Ni _{0.13} Co _{0.13} O _{2+δ-x} F _x	2×10 ⁻¹⁰ -8×10 ⁻¹⁰	5
O3-LiNi _{0.90} Co _{0.07} Mg _{0.03} O ₂	4×10 ⁻¹⁰	6
O3-Li _{1.5} [Mn _{0.75} Ni _{0.15} Co _{0.10}]O _{2+δ}	10 ⁻¹⁰	7
P2-Na _{2/3} Ni _{0.25} Mg _{0.083} Mn _{0.55} Ti _{0.117} O ₂	0.3×10 ⁻¹¹ -0.75×10 ⁻¹¹	8
P2-Na _{2/3} Ni _{1/3} Mn _{5/9} Al _{1/9} O ₂ /RGO	1.54×10 ⁻¹¹	9
P2-Na _{0.65} Li _{0.08} Cu _{0.08} Fe _{0.24} Mn _{0.6} O ₂	10 ⁻¹²	10
P2-Na _{0.66} Ni _{0.26} Zn _{0.07} Mn _{0.67} O ₂ /0.06ZnO	10 ⁻¹²	11
P2-Na _x Fe _{1/2} Mn _{1/2} O ₂	2.0×10 ⁻¹³	12
P2-Na _{0.6} Mn _{0.7} Ni _{0.3} O _{1.95} F _{0.05}	10 ⁻¹⁴	13

- [1] Gan, Q. et al. Polyvinylpyrrolidone-induced uniform surface-conductive polymer coating endows Ni-rich $\text{LiNi}_{0.8}\text{Co}_{0.1}\text{Mn}_{0.1}\text{O}_2$ with enhanced cyclability for lithium-ion batteries. *ACS Appl. Mater. Interfaces*. **11**, 12594-12604 (2019).
- [2] Huang, Y. et al. Tellurium surface doping to enhance the structural stability and electrochemical performance of layered Ni-rich cathodes. *ACS Appl. Mater. Interfaces*. **11**, 40022-40033 (2019).
- [3] Wang, Q. et al. Origin of structural evolution in capacity degradation for overcharged NMC622 via operando coupled investigation. *ACS Appl. Mater. Interfaces*. **9**, 24731-24742 (2017).
- [4] Zhou, P. et al. Stable layered Ni-rich $\text{LiNi}_{0.9}\text{Co}_{0.07}\text{Al}_{0.03}\text{O}_2$ microspheres assembled with nanoparticles as high-performance cathode materials for lithium-ion batteries. *J. Mater. Chem. A*. **5**, 2724-2731 (2017).
- [5] Cao, S. et al. Suppressing the voltage decay based on a distinct stacking sequence of oxygen atoms for Li-rich cathode materials. *ACS Appl. Mater. Interfaces*. **13**, 17639-17648 (2021).
- [6] Zhang, Y. et al. $\text{LiNi}_{0.90}\text{Co}_{0.07}\text{Mg}_{0.03}\text{O}_2$ cathode materials with Mg-concentration gradient for rechargeable lithium-ion batteries. *J. Mater. Chem. A*. **7**, 20958-20964 (2019).
- [7] Yu, R. et al. Self-assembly synthesis and electrochemical performance of $\text{Li}_{1.5}\text{Mn}_{0.75}\text{Ni}_{0.15}\text{Co}_{0.10}\text{O}_{2+\delta}$ microspheres with multilayer shells. *J. Mater. Chem. A*. **3**, 3120-3129 (2015).
- [8] Huang Y. et al. Vitalization of P2- $\text{Na}_{2/3}\text{Ni}_{1/3}\text{Mn}_{2/3}\text{O}_2$ at high-voltage cyclability via combined structural modulation for sodium-ion batteries. *Energy Stor. Mater.* **29**, 182-189 (2020).
- [9] Zhang, X. et al. P2- $\text{Na}_{2/3}\text{Ni}_{1/3}\text{Mn}_{5/9}\text{Al}_{1/9}\text{O}_2$ microparticles as superior cathode material for sodium-ion batteries: enhanced properties and mechanism via graphene connection. *ACS Appl. Mater. Interfaces*. **8**, 20650-20659 (2016).
- [10] Qi, R. et al. A highly-stable layered Fe/Mn-based cathode with ultralow strain for advanced sodium-ion batteries. *Nano Energy*. **88**, 106206 (2021).
- [11] Zhang, F. et al. Stabilizing P2-type Ni-Mn oxides as high-voltage cathodes by a doping-integrated coating strategy based on zinc for sodium-ion batteries. *ACS Appl. Mater. Interfaces*. **13**, 40695-40704 (2021).
- [12] Li, M. et al. Eutectic synthesis of the P2-type $\text{Na}_x\text{Fe}_{1/2}\text{Mn}_{1/2}\text{O}_2$ cathode with improved cell design for sodium-ion batteries. *ACS Appl. Mater. Interfaces*. **12**, 23951-23958 (2020).
- [13] Kang, W. et al. Tunable electrochemical activity of P2- $\text{Na}_{0.6}\text{Mn}_{0.7}\text{Ni}_{0.3}\text{O}_{2-x}\text{F}_x$ microspheres as high-rate cathodes for high-performance sodium ion batteries. *ACS Appl. Mater. Interfaces*. **13**, 15333-15343 (2021).

Comment 10: Page 19: A constant charge/discharge current of 20 mA g⁻¹ was applied by using a MACCOR system between 0.02 V and 2.0 V. This is not the correct voltage range.

Response 10: Thanks for the careful check. We apologize for the typos. The voltage range for in situ XRD measurement should be 2.0-4.3 V, consistent with the electrochemical testing. We have corrected it in the revised manuscript.

Comment 11: charge/discharge current of 20 mA g⁻¹ (what C rate does this correspond to, and which is the theoretical capacity of all the investigated materials? Please add info.

Response 11: Thanks for these comments.

In our work, 1 C is set to 120 mA g⁻¹. 1 C=120 mA g⁻¹ is commonly used in literatures of P2-type layered cathode materials.[1-6] The charge/discharge current of 20 mA g⁻¹ corresponds to 0.17 C.

The theoretical capacities of all the investigated materials can be calculated based on the following formula:

$$\begin{aligned}C_t (\text{mA h g}^{-1}) &= \frac{n \times F (\text{C mol}^{-1})}{M_w (\text{g mol}^{-1})} \\&= \frac{n \times 96485 (\text{C})}{M_w (\text{g})} = \frac{n \times 96485 (\text{A s})}{M_w (\text{g})} \\&= \frac{n \times 96485 \times 1000 / 3600 (\text{mA h})}{M_w (\text{g})} \\&= \frac{26801 \times n}{M_w} (\text{mA h g}^{-1})\end{aligned}$$

Where F is the Faraday constant (96 485 C mol⁻¹), n is the transferred electron number per molecule (n=0.62 in this system), M_w is the molar mass of the P2-Na_{0.62}Mn_{0.67}Ni_{0.23}Cu_{0.05}Mg_{0.09-2y}Ti_yO₂ materials.

The theoretical capacities of Na_{0.62}Mn_{0.67}Ni_{0.23}Cu_{0.05}Mg_{0.09-2y}Ti_yO₂ samples are list in Table R9. However, please keep in mind that the theoretical capacity must be aligned with the working voltage range.

Table R9 Theoretical capacities of P2- $\text{Na}_{0.62}\text{Mn}_{0.67}\text{Ni}_{0.23}\text{Cu}_{0.05}\text{Mg}_{0.09-2y}\text{Ti}_y\text{O}_2$ samples prepared in our work.

Samples	Theoretical capacity (mAh g ⁻¹)
CuMgTi-533	163.05
CuMgTi-552	163.04
CuMgTi-571	163.03
NaMNO ₂	158.50

[1] Xiao, Y. et al. A stable layered oxide cathode material for high-performance sodium-ion battery. *Adv. Energy Mater.* 1803978 (2019).

[2] Shen, Q. et al. Dual-strategy of cation-doping and nanoengineering enables fast and stable sodium-ion storage in a novel Fe/Mn-based layered oxide cathode. *Adv. Sci.* **7**, 2002199 (2020).

[3] Liu, X. et al. Al and Fe-containing Mn-based layered cathode with controlled vacancies for high-rate sodium ion batteries. *Nano Energy* **76**, 104997 (2020).

[4] Fang, Y., Yu, X. & Lou, X. A Practical high-energy cathode for sodium-ion batteries based on uniform P2- $\text{Na}_{0.7}\text{CoO}_2$ microspheres. *Angew. Chem. Int. Ed.* **56**, 1-6 (2017).

[5] Sun, C. et al. Construction of the $\text{Na}_{0.92}\text{Li}_{0.40}\text{Ni}_{0.73}\text{Mn}_{0.24}\text{Co}_{0.12}\text{O}_2$ sodium-ion cathode with balanced high-power/energy-densities. *Energy Storage Materials* **27**, 252-260 (2020).

[6] Zuo, W. et al. The stability of P2-layered sodium transition metal oxides in ambient atmospheres. *Nat. Commun.* **11**, 3544 (2020).

Comment 12: Add amount electrolyte used in each electrochemical test.

Response 12: Thanks for the careful check on the experimental information. The amount of electrolyte used in the coin cells was controlled at around 280 μL , which has been added in the experimental section of revised manuscript.

Comment 13: High purity Na-metal. What is the source? Is that a metal foil or have authors produced a metal foil from a precursor (e.g. metal ingots covered by oxide, or metal cubes under mineral oil?) Please add info.

Response 13: Na metal foil was used as anode for half cells, which was made from Na chunks under mineral oil, which was purchased from Alfa Aesar Co. Na chunk was then pressed into a thin sheet inside glove box, which was further punched into 16 mm diameter plates and used as anode.

We have added the relevant information in the experimental section of revised manuscript.

Reviewer# 2

This manuscript aims to investigate the effect of high entropy that was induced in the materials through multiple cations doping, on the preferential crystal growth in layered oxides, which in turn could affect the electrochemical performance of these oxides used as cathode in Na cells. This topic is very recent, and more work should be done to ascertain the properties and exploit the applications of high entropy materials. Indeed, I have found this work interesting to read and well conducted. In my opinion it can be published in Nat. Comm. after few corrections. There's a couple of things I would ask the author to amend:

General response: We thank the reviewer for the recommendation and valuable comments. We have provided a point-by-point response to your comments:

Comment 1: On pg. 8, lines 172-175 there is a comment concerning the effect of entropy on the growth rate of different crystal facets, that to the best of the authors' knowledge have not been exploited yet. I'm not against the consideration itself; it looks reasonable to me to relate those properties, but surface energies and thermodynamics of crystal growth have long been studied so I'm not sure it would be right to say "have not been exploited" even with a premise. This sentence rose my curiosity and I did a quick search. I can suggest two references as a starting point that might give some clue, i) R. Tran et. al, Surface energies of elemental crystals, Scientific Data, DOI: 10.1038/sdata.2016.80; ii) Talat S. Rahman, Surface Thermodynamics and Vibrational Entropy, Springer Handbook of Surface Science. Therefore, I am asking the authors, if possible, to find another way to express their uncertainty on studies relating entropy and growth rate, because I find that statement too much surprising considering how old is the topic for crystallographers and chemical physicists. Maybe readers could draw their own conclusions just saying "these results indicate that (configurational) entropy variation may reasonably alter the surface energy of crystal planes influencing the growth rate of different facets, which would need additional investigations beyond the primary scope of this work" or similar, in a way that the authors consider appropriate.

Response 1: Thank you very much for the valuable comment and suggestion. According to your suggestion, we have read many articles about entropy and surface free energy,[1-7] and found that research on relation between configurational entropy and surface free energy, surface structure in field of surface science indeed started early (*Science*, **253**, 171-173 (1991)). However, tuning the surface structure of electrode materials by controlling configurational entropy has not been reported so far. Thus, the statement on pg. 8 lines 172-175 should be confined in field of electrode materials and has been modified as “These results indicate that (configurational) entropy variation may reasonably alter the surface energy of crystal planes influencing the growth rate of different facets. To the best of our knowledge, the effect of configurational entropy on surface structure of electrode materials have been barely reported and required more comprehensive study.”

[1] Snyder, E. et al. Atomic force microscope studies of fullerene films: highly stable C₆₀ fcc (311) free surfaces. *Science*, **253**, 171-173 (1991).

[2] Romanyuk, O. et al. Stabilization of semiconductor surface reconstructions by configurational entropy. *Phys. Rev. B*. **82**, 125315 (2010).

[3] Thomas, J., Ven, A. & Millunchick, J. Considerations for surface reconstruction stability prediction on GaAs(001). *Phys. Rev. B*. **87**, 075320 (2013).

[4] Wang, Z., Zhou, W., Li, J. & Wang, J. On the roughening transition of solid/liquid interface in multicomponent alloys. *J. Cryst. Growth*. **502**, 30-34 (2018).

[5] Kapur, S., Prasad, M., Crocker, J. & Sinno, T. Role of configurational entropy in the thermodynamics of clusters of point defects in crystalline solids. *Phys. Rev. B*. **72**, 014119 (2005).

[6] Capdevila-Cortada M. & López N. Entropic contributions enhance polarity compensation for CeO₂ (100) surfaces. *Nat. Mater*. **16**, 328-334 (2017).

[7] Borg, M. et al. Density of configurational states from first principles calculations: the phase diagram of Al-Na surface alloys. *Chemphyschem*. **6**, 1923-1928 (2005).

Comment 2: In this work I've found really a lot of information including the study of the high temperature structural changes by using in situ XRD. All considered, I think that there is no need to discuss samples such as Na_{0.62}Mn_{0.67}Ni_{0.21}Cu_{0.05}Mg_{0.016}Mg_{0.016}Zn_{0.016}Sn_{0.016}Y_{0.016}O₂ and

$\text{Na}_{0.62}\text{Mn}_{0.67}\text{Ni}_{0.21}\text{Cu}_{0.05}\text{Mg}_{0.016}\text{Mg}_{0.016}\text{Zn}_{0.016}\text{Sn}_{0.016}\text{Zr}_{0.016}\text{O}_2$. This is kind of extra information, not really useful and not really interesting because the authors have explained already well their points with the CuMgTi-5ab samples. I would remove those samples and the relative figures and discussions from the main article and the supplementary information to make the reading lighter and fluid.

Response 2: We thank the reviewer for this suggestion. Nevertheless, these two materials are deemed as an extension work of high-entropy cathode compared to CuMnTi-571. We therefore would like to keep it in our work.

Following your suggestion, we have moved all of their data in supplementary information.

Comment 3: The cycling performance shows good results in terms of capacity for Na cells, but the coulombic efficiency has not been described.

Response 3: Thanks for the suggestion. The coulombic efficiencies of CuMgTi-571 and NaMNO₂ cathodes during cycling were added into the revised Fig. 4a, 4e and 4f. And corresponding discussion was added in the revised manuscript.

Fig. 4a compares the cycle performance of CuMgTi-571 with higher entropy and NaMNO₂ with much lower entropy at 0.1 C (1C=120 mA g⁻¹). As shown, CuMgTi-571 could deliver an initial discharge capacity of 148.2 mAh g⁻¹, and can still maintain 132.9 mAh g⁻¹ after 100 cycles, resulting in a high capacity retention of 89.6%. Although NaMNO₂ delivers a comparable first discharge capacity (150.5 mAh g⁻¹) with CuMgTi-571, the discharge capacity of NaMNO₂ decreases fast and remains only 60.2% (90.6 mAh g⁻¹) of its initial capacity after 50 cycles. Moreover, the average coulombic efficiency of CuMgTi-571 achieves 98.0% compared to 95.4% of NaMNO₂, suggesting the fast dynamics and high reversibility of CuMgTi-571.

These detailed analysis on cycle performances and coulombic efficiencies of CuMgTi-571 and NaMNO₂ have been added in the revised manuscript.

Fig. 4 Electrochemical performance of $\text{Na}_{0.62}\text{Mn}_{0.67}\text{Ni}_{0.23}\text{Cu}_{0.05}\text{Mg}_{0.09-2y}\text{Ti}_y\text{O}_2$ and NaMNO_2 samples in the voltage window of 2.0-4.3 V: (a) cycle performances and coulombic efficiencies of CuMgTi-571 and NaMNO₂ at 0.1 C; (b, c) normalized charge/discharge profiles at 0.1 C at different cycle numbers; (d) rate capability; (e) cycle performances at 1.0 C for 500 cycles; (f) long-term cycling behavior at high rate of 10 C. (g) Cycle number, capacity retention, rate properties, current rates and entropy of CuMgTi-571 in this work compared with other reported P2 layered cathodes.

Comment 4: I think ref 22 has been given a wrong article title

Response 4: Thank you very much for the careful check. We apologize for citing the incorrect reference. Ref 22 has been corrected in the revised manuscript.

22. Wang, Q. et al. Multi-anionic and -cationic compounds: new high entropy materials for advanced Li-ion batteries. *Energy Environ. Sci.* 12, 2433-2442 (2019).

Comment 5: I don't know if the session "Discussion" included the Conclusion since I can't find this latter.

Response 5: Discussion includes the conclusion, which is the last paragraph of the article, which is one of the template formats of Nature Communications.

We have added “in summary” at the beginning of the last paragraph to specify our conclusion.

Comment 6: I suggest including standard deviation at least for lattice parameters, cell V, obtained by Rietveld refinement.

Response 6: Thank you for your suggestion. The standard deviation for lattice parameter and cell V were added in the revised supplementary Tables.

Supplementary Table 2. Crystallographic details of the CuMgTi-533 obtained from Rietveld analysis.

CuMgTi-533					
Space group = $P6_3/mmc$					
$a = b = 2.8854 \pm 0.0002 \text{ \AA}$, $c = 11.1884 \pm 0.0009 \text{ \AA}$, $V = 93.15 \pm 0.01 \text{ \AA}^3$, $R_{wp}=9.00$					
atom	site	x	y	z	occupancy
Na	2d	1/3	2/3	3/4	0.4
Na	2b	0	0	1/4	0.22
Mn	2a	0	0	0	0.65
Ni	2a	0	0	0	0.23
Cu	2a	0	0	0	0.05
Mg	2a	0	0	0	0.03
Ti	2a	0	0	0	0.02
O	4f	1/3	2/3	0.06710	1

Supplementary Table 3 Crystallographic details of the CuMgTi-552 obtained from Rietveld analysis.

CuMgTi-552					
Space group = $P6_3/mmc$					
$a = b = 2.8881 \pm 0.0001 \text{ \AA}$, $c = 11.1822 \pm 0.0006 \text{ \AA}$, $V = 93.277 \pm 0.006 \text{ \AA}^3$, $R_{wp}=8.707$					
atom	site	x	y	z	occupancy
Na	2d	1/3	2/3	3/4	0.4
Na	2b	0	0	1/4	0.22
Mn	2a	0	0	0	0.69
Ni	2a	0	0	0	0.25
Cu	2a	0	0	0	0.05
Mg	2a	0	0	0	0.05
Ti	2a	0	0	0	0.02
O	4f	1/3	2/3	0.06988	1

Supplementary Table 4 Crystallographic details of the CuMgTi-571 obtained from Rietveld analysis.

CuMgTi-571					
Space group = $P6_3/mmc$					
$a = b = 2.8886 \pm 0.0002 \text{ \AA}$, $c = 11.1819 \pm 0.0007 \text{ \AA}$, $V = 93.303 \pm 0.008 \text{ \AA}^3$, $R_{wp}=9.805$					
atom	site	x	y	z	occupancy
Na	2d	1/3	2/3	3/4	0.4
Na	2b	0	0	1/4	0.22
Mn	2a	0	0	0	0.67
Ni	2a	0	0	0	0.24
Cu	2a	0	0	0	0.05
Mg	2a	0	0	0	0.07
Ti	2a	0	0	0	0.01
O	4f	1/3	2/3	0.06690	1

Supplementary Table 5 Crystallographic details of the NaMNO₂ obtained from Rietveld analysis.

NaMNO ₂					
Space group = P6₃/mmc					
a = b = 2.8824 ± 0.0002 Å, c = 11.168 ± 0.001 Å, V = 92.79 ± 0.013 Å ³ , R _{wp} = 9.96					
atom	site	x	y	z	occupancy
Na	2d	1/3	2/3	3/4	0.4
Na	2b	0	0	1/4	0.22
Mn	2a	0	0	0	0.66
Ni	2a	0	0	0	0.35
O	4f	1/3	2/3	0.07021	1

REVIEWERS' COMMENTS

Reviewer #1 (Remarks to the Author):

Authors have comprehensively revised the manuscript and have carefully addressed all the questions raised by the reviewers.

Important aspects in terms of novelty have been clarified, as well as other important concepts associated to structural properties and entropy calculation of the investigated materials.

Accordingly, the manuscript can be accepted in its current version.

Reviewer #2 (Remarks to the Author):

The authors did put great effort into the revision of their manuscript mostly to strengthen the novelty of the research results and also to clarify some aspects related to cations modulation on the interlayer spacing and configurational entropy. After this revision I think that the manuscript can be accepted for publication.